# Preventing depression in high-income countries—A systematic review of studies evaluating change in social determinants

Mary Nicolaou[1,2]*, Laura S. Shields-Zeeman[3,4], Junus M. van der Wal[1,2,5], Karien Stronks[1,2]

1 Department of Public and Occupational Health, Amsterdam UMC, University of Amsterdam, Amsterdam, The Netherlands, 2 Centre for Urban Mental Health, University of Amsterdam, Amsterdam, The Netherlands, 3 Netherlands Institute of Mental Health and Addiction (Trimbos Institute), Utrecht, The Netherlands, 4 Department of Interdisciplinary Social Science, Utrecht University, Utrecht, The Netherlands, 5 Department of Psychiatry, Amsterdam UMC, University of Amsterdam, Amsterdam, The Netherlands

* m.nicolaou@amsterdamumc.nl

**Editor:** katsuya oi, Northern Arizona University, UNITED STATES OF AMERICA

## Abstract

We conducted a systematic review to examine whether changes in social determinants can contribute to the prevention of depression, in order to provide input for policy development and to highlight research gaps. Social determinants were defined as the structural conditions in which people live that shape their health and were categorized according to whether they pertained to societal arrangements, material resources distributed through these arrangements, or social resources that follow from interactions between people. To capture all relevant evidence we included studies that measured depressive disorders, depressive symptoms, psychological distress, mental health and prescription rates of antidepressants. We searched three databases (Medline, Embase and Psychinfo) from their inception till December 2022 and supplemented our search by reference and citation searching of the included studies. Studies were synthesized qualitatively and we used the Validity Assessment tool for econometric studies to assess study quality. Prospero submission number CRD42021236132 A total of 31,103 titles were identified, 135 studies met our inclusion criteria. The majority of studies were conducted in the United States (n = 45) or the United Kingdom (n = 39). Studies used longitudinal data (n = 61); repeated cross-sectional data (n = 20); or evaluated an intervention study (n = 7). Study designs included natural experiments (n = 19), while some used propensity score matching to construct a quasi-experiment (n = 11). Analysis methods included difference-in-difference approaches (n = 30) or regression analysis in varying forms. We found evidence that strategies that promote paid employment and parental leave policies can reduce risk of depression whereas reduced entitlements to social welfare (particularly when accompanied by obligations to enter employment), loss of income,

**Data availability statement:** All relevant data are within the article and its supporting information files.

**Funding:** The author(s) received no specific funding for this work.

**Competing interests:** The authors have declared that no competing interests exist.

instability of housing and collective insecurity increase depression risk. A number of studies examined moderation by gender, age category or ethnicity and of these gender was the most commonly observed moderator. Few studies tested underlying causal mechanisms with formal mediation analyses. These studies provide important indications of how intervening on social determinants of health can shape risk for depression. However, the included studies do not fully capture the complexity of the relationships between determinants and the mechanisms driving them. Future studies could take this into account, for instance by using systems approaches.

## Introduction

Globally, depression is one of the leading causes of burden of disease worldwide (ranked 13th among all illnesses) with 279.6 million people living with depression in 2019. [1]. In high-income countries, prevention interventions have mainly focused on the screening, identification and treatment of those at high-risk of developing depression across the life course, particularly in children and young people, young mothers, people of working age and older people [2,3]. Prior research indicates that selective and indicated prevention interventions are important in reducing depressive symptoms in young adults [4] and in preventing the incidence of a (new) depressive episode in the population in general [5]. Although such interventions can have an impact, a recent meta-analysis showed that incidence was only reduced by 20%, leaving 80% unaccounted for [5]. One reason for this may be that interventions target individual-level behaviours, cognitions and strategies, while the social and economic drivers of depression are rarely targeted.

There is increased acknowledgement of the role of social determinants of health in depression [6,7]. Social determinants of health are defined as "non-medical factors affecting health outcomes…the conditions in which people are born, grow, work, live and age and the wider set of forces and systems shaping the conditions of daily life" [8]. These include factors such as the built environment (e.g., blue/green space) [9], institutional structures (e.g., policy, healthcare services) [6,10], and social conditions (e.g., social cohesion, neighbourhood crime) [11,12]. The importance of the impact of social determinants is evidenced by the inequalities in depression rates between population sub-groups, i.e., the prevalence of depression is often higher in groups such as ethnic minorities and groups with a low socioeconomic status or with high deprivation, in which exposure to unfavourable social conditions is higher [13–17].

The evidence-base for the role and impact of social determinants on mental health outcomes, including depression is growing [10,11,18–21]. However, in existing studies it is often not clear whether a *change* in exposure to social determinants will also lead to a *lower risk* of depression. Establishing causality is fraught with difficulty and the increased application of counterfactual methodologies to this field has helped to clarify causality [22]. While there have been reviews of the evidence for changes in mental health and depression in relation to changes in income [21] and social security eligibility [10], an overview of the impact of a broad range of social

determinants is missing. This is necessary to establish which types of (policy) interventions have the potential to improve population mental health, in particular depression, as well as to inform policy makers on possible adverse mental health effects of certain policies.

The aim of this review is to examine whether change in a diverse set of social determinants can contribute to depression prevention. The results will provide insights for policy development and highlight research gaps.

## Methods

We conducted a systematic review to identify social determinants for which there is evidence for an impact on the risk of depression or depressive symptoms following changes in those determinants. The review protocol was registered in the International Prospective Register of Systematic Review (Prospero), submission number CRD42021236132. Available at https://www.crd.york.ac.uk/prospero/display_record.php?ID=CRD42021236132

Social determinants are the structural conditions in which people live that shape their health. We considered a determinant as 'social if it is the outcome of an interaction between individuals [23]. This can take the form of:

• *societal arrangements* aimed to facilitate individuals living together in society, including the *physical infrastructure*, e.g., public transport, green/blue space and *institutions*, i.e., social norms, rules, structures impacting individual behaviour, such as the social welfare system;

• *material resources distributed* within societal arrangements, such as social benefits or income. Material resources not distributed through a societal arrangement, such as an increase of income due to a lottery, are excluded from this review.

• *social resources that follow from the direct interaction of people* in a certain context, e.g., social disorder, social participation and social capital.

### Outcome measure

Our main outcome of interest was depressive disorders and depressive symptoms and we included studies that measured these outcomes using either a clinical interview or a validated screening tool such as the Center for Epidemiologic Studies Depression Scale (CES-D). In order to capture all relevant evidence, we also included studies that measured psychological distress or general mental health status such as the General Health Questionnaire (GHQ), the 12-Item and 36-Item Short Form Health Survey (SF-12/SF-36), the Mental Health Inventory 5 (MHI-5 scale), or the Kessler Psychological Distress Scale (K-6/K-10), as these questionnaires often have an item related to depressive symptoms or indicate poor mental health. Additionally, we included studies that evaluated prescription rates of antidepressant medication or other health care use indicators as a proxy outcome for the presence of depression. In the results section we describe outcomes measures as:

• Mental health or psychological distress measured by instruments such as K-6, K-10, MHI-5, GHQ, SF-12.

• Depressive symptoms based on self-reported questionnaires such as Patient Health Questionnaire (PHQ), CES-D, Beck Depression Inventory, Hopkins Symptom Inventory.

• Depression based on diagnostic interviews.

• Mental healthcare use, including prescriptions, visits to mental health care services.

### Types of studies included

• Original studies that have evaluated changes in outcome measures and change in social determinants over time (two or more moments): repeated cross-sectional, longitudinal and (quasi)experimental studies including randomized controlled trials (RCTs).

• Studies set in high-income countries as defined by the World Bank [24].

• Studies that included adolescents (12–18 years) and adults aged 18 years and older.

• Studies with a sample of participants living in non-institutionalized settings.

**Types of studies excluded**

Studies were excluded if:

• The outcome was measured at only one moment in time, such as cross-sectional studies.

• The exposure was unrelated to a social determinant, e.g., screening for depression; stress management interventions; healthcare referral schemes.

• The study population was focused on children younger than 12 years.

• Studies in specific clinical populations or patient groups, e.g., studies among persons with diabetes, heart disease etc.

• Simulation or computational modelling studies.

• Non-human/laboratory-based studies.

• Studies set in low and middle-income countries.

**Search strategy**

We searched for peer-reviewed papers published in English up to December 2022. Our search terms were informed by a previous scoping review [6]; from a previous review on a similar topic [10]; and refined based on pilot searches. Three bibliographic databases were examined; Medline (including MeSH terms), Embase, and PsycINFO. The database searches were supplemented by reference and citation searching of the included studies and previously published reviews on similar topics. Search terms included three major concepts comprising a range of social determinants such as natural, structural and social environments, social and economic policy domains such as social welfare, housing, employment; outcomes (mental health and well-being, depression), and methods (quantitative). The full search strategy is included in the appendix "S1 Data".

**Title and abstract screening**

Papers identified by the search strategy were uploaded in Rayyan [25] for screening. MN, LSZ and KS screened titles and abstracts of identified studies for relevance, according to the review inclusion criteria. In cases of doubt the rationale for inclusion/exclusion were discussed to reach consensus. Due to the large number of papers identified, JvdW and MN additionally screened a random set of 10% of the included papers as a check.

**Full-text reviews and data extraction**

The retrieved full texts were reviewed by MN and KS for inclusion. An extraction table was developed to contain general information and study characteristics (author, year of publication, country, study design, sample size), population characteristics (gender, age, ethnicity), outcome measure as described by the authors (e.g., mental health; psychological distress; depressive symptoms; depression; etc), type of social determinant studied (income, social cohesion etc.). Reasons for excluded papers were documented and are available in appendix "S2 Data".

Studies were summarised narratively due to the wide variety of exposures and outcomes evaluated. As stated in our registered protocol, we originally planned to use the Dahlgren and Whitehead model of social determinants of health [26]

as a framework to classify identified determinants at the individual, community, or societal level; however in practice it was difficult to apply this framework. For example, the introduction of a minimum wage can be categorised as a societal-level intervention but has an impact on individual-level income in those to whom it applies [27]. We therefore grouped studies based on the aforementioned categorisation building on Hahn [23]: societal arrangements; material resources; and social resources.

Outcomes were considered as statistically significant if they were below the conventional p-value cut-off of 0.05. We reported results of sub-group analyses and potential mediators when included by the original studies to understand potential differential effects by population characteristics or underlying mechanisms driving the main effect being studied.

## Quality assessment

Evaluation of study quality was piloted using the Quality Assessment Tool For Quantitative Studies [28] as well as the ROBINS-I tool by Sterne JAC et al [29] as per our published protocol. However, both tools were inappropriate due to their focus on traditional study methods. We used the Validity Assessment tool for econometric studies by Barr et al. [30], a tool developed for econometric studies, which better suited the studies included in this review. The Validity Assessment includes nine component rating sections (unit of analysis; comparison approach; sample selection; number of time points of data; response bias; exogeneity of policy exposure, confounding, sample size/power and statistical methods). Overall scores ranged from 0 to 27. Consistent with Simpson et al [10], we considered studies scoring 18 or higher as high quality. MN and JvdW evaluated study quality and consulted LSZ in cases where consensus was not reached. Our quality assessment tool did not include scores for the operationalisation of the outcome measures of this review.

## Results

Our search identified 31,103 non-duplicate titles: 30,961 from database searches and 142 from reference and citation searches. Of these, 116 studies (134 reports) met our inclusion criteria and were included in the final review (41 from databases and 93 via citation searching). See Fig 1, PRISMA flow diagram.

### Study characteristics

A full overview of all included studies and their respective characteristics can be found in Tables 1–3. 45 studies were carried out in the USA; 39 in the UK (including England, Scotland and Wales); six in the Netherlands and Australia; five in Canada, Sweden and Spain; three in Japan, South Korea, Greece and Norway; two studies that included multiple European countries, Ireland and France; and one in Denmark, Hong Kong, Finland, New Zealand, Belgium and Germany. Most studies included adults aged 18 years and older but studies that focused on the working population included a slightly younger population aged 15/16 years and older. Eight studies included older adults, three studies (four reports) focused only on adolescents.

Most studies used longitudinal (cohort) data (n = 91); 33 studies used repeated cross-sectional data; seven studies evaluated an intervention study, five of which were randomised and two non-randomised; two studies used repeated cross-sectional data to construct a cohort. Study designs included natural experiments (n = 19), while some used propensity score matching to construct a quasi-experiment (n = 11). Analysis methods included difference-in-difference approaches (n = 30) or regression analysis in varying forms.

Based on our quality assessments, almost all studies scored equal or higher to 18 points (of maximum 27) and were rated 'good' quality (n = 121). Strengths included use of random population samples, inclusion of multiple covariates in models, large sample sizes and appropriate analyses; specifically, recent studies (n = 33) were more likely to employ a difference-in-difference analysis strategy. The most common weaknesses were the limited number of data-points included in the analysis, with most studies including two time points; high risk of bias through low response rates; 31 studies scored low (1 out of possible 3 points) on the 'exogeneity of policy measure' component of the instrument used.

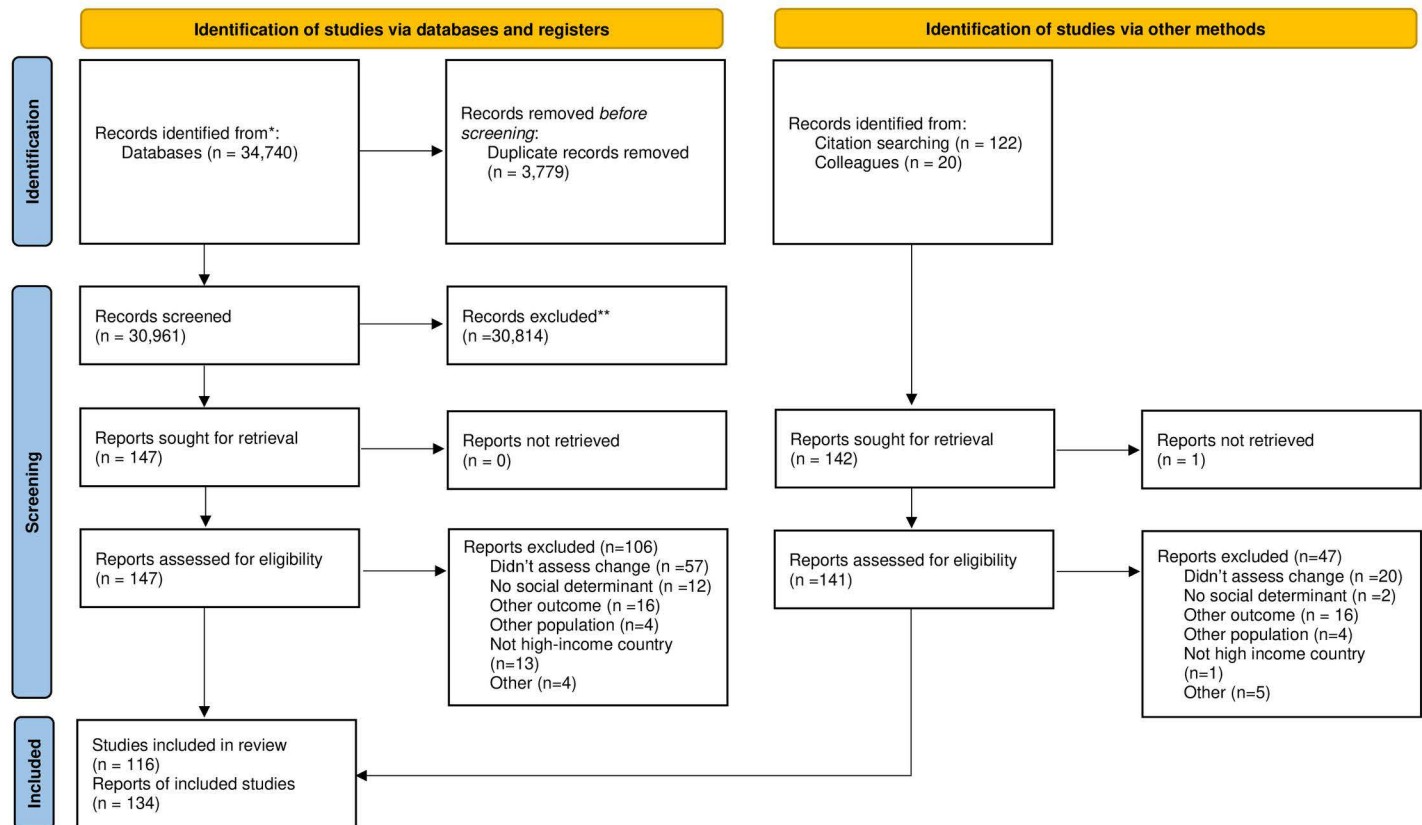

**PRISMA 2020 flow diagram for new systematic reviews which included searches of databases, registers and other sources**

*From:* Page MJ, McKenzie JE, Bossuyt PM, Boutron I, Hoffmann TC, Mulrow CD, et al. The PRISMA 2020 statement: an updated guideline for reporting systematic reviews. BMJ 2021;372:n71. doi: 10.1136/bmj.n71. For more information, visit: http://www.prisma-statement.org/

**Fig 1. PRISMA flow diagram.**

The majority of studies (n = 126) used questionnaires to operationalise outcomes. Mental health and psychological distress were measured using the SF-12/36/MHI-5 (n = 16), GHQ (n = 23), the K-6/K10 (n = 12). Depressive symptoms were measured mostly using the CES-D (n = 26). Other instruments used include the Beck Depression Inventory and the Hopkins Symptom Checklist. Eight studies used short (<12 items) non-validated questions/questionnaires, e.g., mental health questions as part of a broader population heath survey. Only eight studies (nine reports) used a diagnostic clinical interview for depression. Finally, four studies analysed records of antidepressant prescriptions and seven studies used treatment claims or hospital admission data.

Results are classified in Tables 1–3 according to a single social determinant, although some studies analysed several determinants, which is reflected by those studies being mentioned more than once in Table 4.

## Study findings: Societal arrangements

Included studies were sub-divided into welfare reform policies, minimum wage policies, educational policies, public transportation policies and environmental interventions, and are shown in Table 1.

**Welfare reform: Tax credit policies.** Several US studies examined the effect of the expansion of the Earned Income Tax Credit (EITC) on depression symptoms or psychological distress. The EITC, implemented in 1975, was designed

**Table 1. Studies addressing societal arrangements.**

| Reference & country | Intervention | Target population, sample size | Study design | Analysis | Data source | Outcome and measure | Results | Sub-group analysis (moderation) | Mechanisms (mediation) | QA |
|---|---|---|---|---|---|---|---|---|---|---|
| **Welfare reforms: Tax credit benefits (income supplementation)** | | | | | | | | | | |
| Shields-Zeeman et al. 2021. USA (32) | EITC impact on household income | Adults: heads of households and spouses (n = 36262). | Cohort | Instrumental variable (EITC eligibility and size of refund); regression (linear, fixed effects) | Panel Study of Income Dynamics – 1985–2015 (bi-annual data) | Psychological distress; K-6 | Higher income and larger EITC refund associated with lower psychological distress | Effect for single income households; no effect if married, or on basis of income restriction. | – | 26 |
| Spencer et al. 2020. USA (34) | Policies to supplement income: Temporary Assistance for Needy Families (TANF), Minimum Wage (MW), and EITC | Mothers with/without high-school diploma. n = 3545 | Cohort | Difference-in-Difference; regression, fixed effects. | Fragile Families and Child Well-being birth Cohort. Waves 1–4 (birth through 9yrs) | Depression. Composite International Diagnostic Interview-Short Form | No statistically significant effects of all three policies on depression | No difference by ethnicity – AA versus white | – | 23 |
| Collin et al. (2020) USA (33) | EITC – short-term effect immediately after income receipt | Persons receiving EITC – persons with income between $0 -$100,000 (n = 379,603 and 29,808) | Repeated cross-sectional and Cohort | Difference-in-Difference | The National Health Interview Survey (1997–2016). The Panel Study of Income Dynamics 1985–2015 | Psychological distress; K-6 | No effects of the EITC on short-term psychological distress | – | – | 24 |
| Boyd-Swan et al. 2016 USA (31) | Earned Income Tax Credit (EITC) expansion 1990 – wage supplement | Adult women 16–55 years, eligible for EITC (n = 5557) | Cohort | Difference-in-Difference; Regression (OLS) | The National Survey of Families and Households 1987–88 and 1992–94 (2 waves) | Depression. CES-D-11 | Reduction in depression in mothers. | Reductions significant in married mothers. No effect on single mothers | – | 21 |
| **Welfare reforms: Minimum wage legislation** | | | | | | | | | | |
| Maxwell et al. 2022. UK (37) | 2016, 2017 and 2018 UK National Minimum Wage increases | Adults aged 25–64 at the time of interview (2016 n = 803; 2017 n = 1083; 2018 n = 803) | Natural experiment. Longitudinal study | Difference-in-difference using before and after measures for each year studied | Understanding Society survey, representative survey households in the IK. | Mental Health. SF-12 mental health component | No statistically significant difference in mental health at any of the time points. | – | – | 23 |

*(Continued)*

**Table 1.** (Continued)

| Reference & country | Intervention | Target population, sample size | Study design | Analysis | Data source | Outcome and measure | Results | Sub-group analysis (moderation) | Mechanisms (mediation) | QA |
|---|---|---|---|---|---|---|---|---|---|---|
| Busz-kiewicz et al. 2019. USA (38) | State minimum wage (above federal minimum) | working-age adults (25–64 years) (n = 131,430) | Repeated cross-sectional | difference-in-difference-in-differences (DDD) approach; and difference-in-difference. | State-level National Health Interview Survey (NHIS) data linked with state policy and economic indicators, 2008–2015) | Psychological distress: K-6 | No association with psychological distress | No difference in effect based on gender, age, race/ethnicity, employment status. | – | 22 |
| Reeves et al. 2017. UK. (27) | Wage increase – introduction of minimum wage 1999 | Adults aged 22–59yrs. (n = 279). Eligible for wage increase vs those at 100–110% threshold | Repeated cross-sectional | Difference-in-Difference; regression (fixed-effects) | British Household Panel Survey: 1998 (pre) and 1999 (post); | Psychological distress; GHQ-12 | Reduced Psychological distress | – | Financial strain attenuated effect in both intervention and control. | 21 |
| Kronenberg et al. 2017. UK (36) | Introduction of minimum wage 1999 | Low-wage earners eligible for minimum wage, aged ≥18 years (n not specified) | Repeated cross-sectional | Difference-in-difference | British Household Panel Survey 1997–2000 (3 waves) | Psychological distress; GHQ-12 | No association with psychological distress. | – | – | 20 |
| **Welfare reforms: Expanding access health insurance** | | | | | | | | | | |
| Schuster et al. 2022. USA (41) | Expanding health insurance coverage due to the Affordable Care Act in 2014 | Women eligible for ACA (n = 9472) | Repeated cross-sectional | Linear probability models. | 2012–2015 data from the Phase 7 Questionnaire of the Pregnancy Risk Assessment Monitoring System. | Post-partum depression based on 2 questions. | Lower estimated proportion of women reporting post-partum depression after introduction of Affordable Care Act. | – | – | 22 |
| Mukhopad-hyay. 2022. USA (40) | Expansion of Medicaid in context of job loss during covid-19 pandemic | Persons who lost work or had family member who lost work during pandemic. (n = 872,428) | Repeated cross-sectional. Compare states with expansion of Medicaid with non-expansion states. | Difference-in-difference | 27 rounds of the Household Pulse Survey -nationally representative survey. Covering April 2020 to March 2021. | Mental Health. PHQ-4 | Respondents in expansion states were 14.2% less likely to have moderate to severe mental distress following job loss compared to those living in non-expansion states. | – | Access to health-care; financial security; food security | 23 |

*(Continued)*

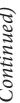

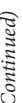

| Reference & country | Intervention | Target population, sample size | Study design | Analysis | Data source | Outcome and measure | Results | Sub-group analysis (moderation) | Mechanisms (mediation) | QA |
|---|---|---|---|---|---|---|---|---|---|---|
| Finkelstein et al. 2012. USA (39) | State-funded health insurance | Low-income adults who signed up for the Medicaid 'lottery' (n = 23,741) | Natural experiment; Cohort | Instrumental variable (lottery in 2008), Linear regression | constructed 'Cohort' using mail survey and linkage with administrative data | Depression. PHQ-2 | Compared to non-lottery winners: ITT and TOT lottery groups had lower depression | Effect stronger on those who took out insurance (TOT group). | – | 21 |
| **Welfare reforms: Work incentives and employment** | | | | | | | | | | |
| Williams. 2021. UK (48) | Universal Credit benefit; restriction of Job-seeker's Allowance in 2012 | Adults in the UK. (n = 100,000) | Repeated cross-sectional; ecological-level. 2010–2014 | Regression; regression, fixed effect | the Quarterly Labour Force Survey | Self-reported anxiety/depression (3 items) | Sanctions associated with increased number of people reporting anxiety and/or depression | | | 23 |
| Williams, (2019). UK (49) | Universal Credit benefit; restriction of Job-seeker's Allowance 2012 | Adults in the UK. | Repeated cross-sectional; ecological-level. 2010–2018 | Regression; regression, fixed effect | Constructed longitudinal dataset: local-level data on JSA sanctions and NHS registered antidepressant prescriptions | Anti-depressant prescribing | JSA sanctions associated with increase in number of prescriptions | | | 22 |
| Wickham et al. 2020. UK (47) | Introduction of Universal unemployment benefit 2013 | Men and women of working age (16–64 years). n = 52,187 | Cohort 2009-2018 | Difference-in-Difference. multivariable mixed effects linear regression | Understanding Society UK Longitudinal Household Panel Study | Psychological distress. GHQ-12. SF-12 mental health component | prevalence of psychological distress increased and mental health decreased | | | 23 |
| Katikireddi et al (2018) UK (46) | Lone Parent Obligations – welfare benefits conditional to seeking work based on youngest child's age - Progressive age cut-off: 2008 12 yrs of ages; 2009 10 yrs; 2010 7 yrs; 2012 5 years. | Lone mothers n = 852 and 619 | Cohort | Difference-in-Difference. individual fixed-effects linear regression model | UK Understanding Society Survey | Mental health. SF-12 mental health component. | Conditional benefits associated with increased risk of poor mental health in the short term. | No difference found on the basis of maternal educational level | | 25 |

*(Continued)*

**Table 1.** (Continued)

| Reference & country | Intervention | Target population, sample size | Study design | Analysis | Data source | Outcome and measure | Results | Sub-group analysis (moderation) | Mechanisms (mediation) | QA |
|---|---|---|---|---|---|---|---|---|---|---|
| Harkness. 2016. UK (43) | Welfare reforms: WFTC and other welfare-to-work strategies (1999). | Mothers of working age (<55yrs) + dependent children <16. (n = unclear) | Cohort; data from 1993–98; 1999–2003; and 2003–08 | Regression (OLS) | British Household Panel Survey | Psychological distress. GHQ-12 cut-off (4) | Reduced risk of psychological distress for single mothers compared to married mothers. | – | – | 22 |
| Curnock et al (2016). UK (45) | Transition to work due to changes in eligibility to disability benefits. (Work Capability Assessments) | Persons receiving out-of-work disability welfare benefits ≥16 years. n = 1145–1497 | Cohort | Difference-in-Difference with propensity score matching. | UK Understanding Society panel survey (2009–2013–4 waves) | Mental Health. SF-12 mental health component | Transitioning from disability benefits to work or unemployment status were both associated with improved mental health. | No health differences in those moving to the 'new' disability welfare scheme | | 22 |
| Barr et al. (2015). UK (44) | Reassessment of disability benefits through Work Capability Assessments (2010). | Working age population 18–64 years. | Natural experiment; Repeated cross-sectional data | linear fixed effects multivariable regression | Quarterly Labour Force Survey. Aggregated data for 149 local authorities in England (2004–2013) | Antidepressant-prescribing rates aggregated at local authority level. Depression or anxiety (single item) | Local areas with greater proportion of population exposed to reassessment had greater increase in antidepressant prescribing and self-reported depression | – | – | 22 |
| Rote & Quadagno. (2011). USA (51) | Introduction of the Personal Responsibility and Work Opportunity Reconciliation Act (PRWORA) 1996 | Women 18–49 yrs and living below the poverty line. (1995 n = 1,913 and 2006 n = 3,666) | Repeated cross-sectional | Regression (OLS) | National Survey of Drug Use and Health. 1995 and 2006. | Depression. 8 items based on Diagnostic and Statistical Manual of Mental Disorders III | welfare recipients experienced more depressive symptoms than other poor women | – | – | 22 |
| Jagannathan et al. 2010. USA (50) | Welfare-to-work legislation (New Jersey Family Development Program) | Adult women (n = 8393) | Intervention with random assignment (1992–1996) | ITT; logit regression | The New Jersey Family Development Program | Clinical depression. Treatment claims to Medicaid programme | Increase odds of depression for experimental group | Hispanic women increased risk; Black women decreased risk | | 20 |

*(Continued)*

| Reference & country | Intervention | Target population, sample size | Study design | Analysis | Data source | Outcome and measure | Results | Sub-group analysis (moderation) | Mechanisms (mediation) | QA |
|---|---|---|---|---|---|---|---|---|---|---|
| Gregg et al. 2009. UK (42) | Welfare reforms to incentivise employment: Working Families Tax Credit (WFTC) and New Deal for Lone Parents (active case management) (1999) | Women of childbearing age. N = 27,370 | Cohort; used data from 1991–7 and 1999–2003 | Difference-in-Difference (single mothers; married mothers; and single childless women) | British Household Panel Survey | Psychological distress. GHQ-12 cut-off (4) | Reduced odds of psychological distress in single mothers compared to both controls. | Effects stronger for mothers that were separated for ≤1 year | – | 24 |
| Morris. 2008. USA (52) | Facilitation of employment - Implementation of different strategies in the context of PRWORA | Welfare recipient parents (n = 6,761) | Intervention; random assignment: emphasis on quick job entry versus personal client attention | Regression (linear and logistic) | multi-site experimental evaluations | Depression scale and cut-off. CES-D | For programs at the highest quartile 'quick job entry' dimension, the program group had a higher level of depression than the control group. Caseworkers' personalised attention had no effect. | The effects of emphasis on quick job entry are stronger for parents with preschool children at baseline than for the total sample. | | **21** |

**Welfare reforms: Parental Leave**

| Reference & country | Intervention | Target population, sample size | Study design | Analysis | Data source | Outcome and measure | Results | Sub-group analysis (moderation) | Mechanisms (mediation) | QA |
|---|---|---|---|---|---|---|---|---|---|---|
| Bütikofer et al. 2021. Norway (61) | Introduction of paid maternity leave (July 1977) on medium and long term impact on maternal health. | Women aged 39–42 years at time of survey and gave birth in 1977. (n = not reported) | Compared the health of eligible mothers who had children immediately before and after July 1, 1977. i.e., before 1 July not entitled to benefit. | Regression discontinuity and difference-in-difference design. | Combine Norwegian birth registry data with national-level survey data of maternal health (1988–2003) | Standardized index for self-reported mental health in the past two weeks. | Improved mental health post-reform. | Larger effects on mothers who experienced complications at delivery, first-time mothers, and low-resource mothers (e.g., single mothers; below-median household income) | Improvements were driven by more time at home after child-birth, not changes in income | 22 |
| Cardenas et al. 2021. USA (53) | Paid paternity leave in relation with first-time parents' depression | Couples expecting first child (n = 72) | Longitudinal study comparing couples based on father taking paternity leave or not. | Repeated measures ANCOVAs to examine relationship between paternity leave and changes in depression, | Longitudinal study of transition into parenthood. | Depressive symptoms. Beck Depression Inventory. | Mothers whose partners took paid paternity leave had smaller prenatal-to-postpartum increases in depressive symptoms. No difference in father's depressive symptoms. | – | – | 19 |

(Continued)

**Table 1.** (Continued)

| Reference & country | Intervention | Target population, sample size | Study design | Analysis | Data source | Outcome and measure | Results | Sub-group analysis (moderation) | Mechanisms (mediation) | QA |
|---|---|---|---|---|---|---|---|---|---|---|
| Irish et al. 2021. USA (55) | Paid family leave California (2004) and New Jersey (2009). | Working adults with children <2 years (n = 28,638) | Natural experiment. Repeated cross-sectional | Difference-in-difference | Annual National Health Interview Survey. 1997 – 2016 waves | Psychological distress. Kessler-6. | Lower psychological distress in both mothers and fathers post policy. | Smaller effect on Black and Hispanic participants | – | 22 |
| Doran et al. 2020. USA (54) | Paid family leave on maternal post-partum depression (California 2004) | Mothers with children aged <12 months (n = 7379) | Natural experiment. Repeated cross-sectional. | difference-in-difference | National Health Interview Survey, 11 waves from 2000–2010 | Psychological distress. Kessler-6 | Access to paid family leave was associated with a decrease in postpartum psychological distress symptoms (27.6% decrease from the pre-treatment mean). | Indications of greater effect on black and Hispanic mothers; younger mothers; non-college educated mothers; foreign-born mothers. | – | 24 |
| Lee and Modrek. 2020. USA (56) | Paid family leave California (2004) | Parents with children <2 years (n=6,690) | Cohort. | Difference-in-difference | Panel Study of Income Dynamics. 1993–2017 waves | Psychological distress. Kessler-6. | Improved psychological distress post-policy. | Effect larger among mothers. | – | 22 |
| Bilgrami et al. 2019. Australia (57) | Introduction of paid parental leave (2011), and Dad and Partner Pay (2013). | Working mothers (n-1480 | Longitudinal study. Pre-reform group is control. | | Annual Household, Income, and Labour Dynamics in Australia survey 2004–2016 | Depression based on SF-36 Mental Component Summary using cut-offs for mild, moderate and severe depression. | Reduced maternal depression post-reform. This was greater in women whose partner had access to the DAPP | significant improvements specifically in first-time mothers and those with employer-paid maternity leave and unpaid leave entitlements | – | 21 |
| Hewitt et al. 2017. Australia (58) | Paid parental leave policy 2011 | Mothers eligible for maternal leave (had had a baby) (n = 2347 pre-PPL and 3268 post-PPL) | Natural experiment. Repeated cross-sectional. | Regression. | Two cross-sectional surveys. pre- and post-PPL – 2009 and 2011 | Mental health. SF-12 | Post-PPL mothers' mental health was significantly better compared to those mothers who gave birth pre-PPL | Effects observed in mothers regardless of employment quality. | – | 19 |
| Beuchert et al. 2016. Denmark (60) | Reform in maternity leave: increase in paid leave in march 2002. | Women who gave birth in period Nov 2001-March 2002 (n = 15449) | Cohort | Instrumental variable based on time of giving birth. | Administrative data sets of whole Danish population. | Hospitalisation with depression; receiving anti-depressants | No significant effects of increasing the length of maternity leave on either mental health outcome. | Did not find evidence of difference on the basis of maternal income or education. | – | 23 |

*(Continued)*

Table 1. (Continued)

| Reference & country | Intervention | Target population, sample size | Study design | Analysis | Data source | Outcome and measure | Results | Sub-group analysis (moderation) | Mechanisms (mediation) | QA |
|---|---|---|---|---|---|---|---|---|---|---|
| Avendano et al. 2015. 7 European countries. (59) | Paid maternity leave policies from 1960 onward. Included countries with a change in policy in the time-span. | Women aged ≥50yrs who had a child when aged 16–25 years. (n=2374) | Cohort with retrospective data on maternity leave | Difference-in-difference with women not eligible for paid leave as control. | SHARE=cross-national panel survey of a representative sample of the European population aged 50+ | Depressive symptoms. EURO-Depression (Euro-D) scale | Paid maternity leave was associated with lower depressive symptoms in older age. | – | – | 23 |
| **Welfare reforms: Pension policies** | | | | | | | | | | |
| Carrino, Glaser and Avendano. 2020 UK (63) | Increase of State Pension Age. | Women aged 60–64 years. (n=3,531) | Cohort | Difference-in-Difference; regression (OLS and linear). SPA eligibility based on age (IV) | Understanding Society Survey. Seven waves 2009–2016 | Psychological distress GHQ-12 and SF-12 Mental Health score. Doctor diagnosed depression. | SPA reform increased the likelihood of psychological distress and poor mental health. Dose response with longer postponement. | Negative health effect confined to women from lower-grade routine occupations | – | 23 |
| Kim et al. 2020; South Korea (64) | Introduction of basic Old-Age Pension 2008 | Older adults in ≥70 yrs (n=760) | Cohort | Difference-in-Difference; multi-level | Korea Welfare Panel Study. 2007-2008 | Depression. CES-D-11 | No significant association between the BOP and depression | – | – | 24 |
| De Grip et al. 2011. Netherlands (62) | Increase of State Pension Age. 2006 | Male public sector workers (n=5,274) | Natural experiment; Cohort | Instrumental variable = birth year; regression (OLS)s | Link survey and administrative information from pension fund. 3 waves starting in year after reform till 2008. | Depression. CES-D-8 score with cut-off ≥4 | Reform associated with increase in depression that persisted 2 years after reform announcement. | Effects stronger for workers who experience a larger income loss, and for married men whose partner has no pension income | – | 20 |
| **Welfare reforms: housing benefits** | | | | | | | | | | |
| Reeves et al. 2016. UK (65) | Housing Benefit Reduction (in 2011) | Persons aged 16–69 years who were renting housing in the private sector (n=179 064) | Natural experiment: Repeated cross-sectional; coarsened exact matching. April 2009–March 2011 and April 2011–March 2013 | Difference-in-Difference; linear probability model. | Annual Population Survey UK | Depressive symptoms. Single questionnaire item. | Post reform: Increase in prevalence of depressive symptoms in HB recipients | Increase in depressive symptoms was greater among HB recipients living in high-impact areas than those in low-impact areas | – | 24 |

(Continued)

**Table 1.** (Continued)

| Reference & country | Intervention | Target population, sample size | Study design | Analysis | Data source | Outcome and measure | Results | Sub-group analysis (moderation) | Mechanisms (mediation) | QA |
|---|---|---|---|---|---|---|---|---|---|---|
| **Welfare reforms: Childcare benefits** | | | | | | | | | | |
| Lebihan en Tangkomo 2018. Canada (66) | Unconditional cash transfer to families (Universal childcare benefit) 2015 | Two-parent families with young children (<6 yrs) (n= | Natural experiment- Comparison between eligible and non-eligible families | Difference-in-Difference | National Longitudinal Survey of Children and Youth (NLSCY) 1994–2008, bi-annual survey and the Survey of Young Children (SYC); | Depressive symptoms; 12 item questionnaire (origin not specified) | Small non-significant improvement in depressive symptoms among eligible families | Modest effect based on low parental education level and for girl children. | | 25 |
| **Public transport policies** | | | | | | | | | | |
| Reinhard et al. 2018. UK (67) | Use of Public Transport due to eligibility for free public transport. | Adults aged ≥50 years (n=18453) | Natural experiment; Cohort | Instrumental variable (eligibility for free bus-pass); regression (2SLS) | the English Longitudinal Study of Ageing | Depressive symptoms. CES-D-8 | Using public transport associated with reduced depressive symptoms. | – | – | 23 |
| **Educational policies** | | | | | | | | | | |
| Hamad et al. (2019) USA (68) | Educational achievement. | adults n = 30,853 and 44,732 | Natural experiment; Cohort and repeated cross-sectional. | Instrumental variable (state-level schooling law); regression (OLS) | 1992–2012 waves Health and Retirement Study and NHANES (1971–2012) | Depression. CES-D-8 – cut-off>3 | Higher education associated with lower depression | – | – | 20 |
| **Environmental exposure** | | | | | | | | | | |
| Wang et al. 2022. USA (69) | Increase in aircraft noise exposure due to altered flight paths in 2012 | Medicaid members residing in each of the two neighbourhoods at two points in time | Natural experiment. Cohort. | Difference-in-Difference | New York City Medicaid claims. 2009–2011 and 2013–2016. Exposure based on registered and measured noise-level data. | Depression. International Classification for Disease revision codes. (ICD-9=311or ICD-10=F33) and mood disorder (CCS=657) | Depression diagnoses increased for the age groups 18–44 and 45–64. Broader mood disorder diagnoses, increased in the 45–65 age group | Differences by age group. | – | 22 |
| Li et al. 2020, Netherlands (70) | Environmental noise control policies – reduce aircraft noise level. | Older population in Amsterdam. (n=1746). | Natural experiment. Cohort | Difference-in-Difference | the Longitudinal Aging Study Amsterdam (LASA.) linked to Environmental data | Depressive symptoms. CES-D-20 | No change in depressive symptoms. But estimates suggested that the policy did not lead to reduction in noise levels in treatment relative to control areas | – | – | 24 |

Table 2. Studies addressing material resources.

| Reference & country | Intervention | Target population, sample size | Study design | Analysis | Data source | Outcome and measure | Results | Sub-group analysis (moderation) | Mechanisms (mediation) | QAT |
|---|---|---|---|---|---|---|---|---|---|---|
| **Income supplement** | | | | | | | | | | |
| Courtin et al. 2018. USA (73) | Conditional cash transfers (NYC–Family Rewards) | 4,749 families: 2,377 intervention and 2,372 control | RCT | Regression (OLS) with ITT | Participants in the Family Rewards Experiment – condition that money spent on education, preventive health care, and parental employment | Psychological distress; K-10 | No effect on psychological distress | | | 22 |
| Costello et al. 2010. USA (71) | Fixed income supplement – moving out of poverty. | Follow up of children now aged 21 yrs (n=1185) | Natural experiment (opening casino in 1996); Cohort | GEE models | The Great Smoky Mountains Study follow up 2006 | Depression/anxiety. DSM-IV interview. | No significant effect | No effect of age-Cohort on outcome | | 20 |
| Costello et al. 2003. USA (72) | Fixed income supplement – moving out of poverty. | children aged 9, 11 & 13 yrs at intake (n=1420) | Natural experiment (opening casino in 1996); Cohort | GEE models | The Great Smoky Mountains Study 1993-2000 | Depression/anxiety. DSM-IV interview. | No significant effect | | Indications that adequate parental supervision was a mediator. | 19 |
| **Income volatility** | | | | | | | | | | |
| Thomson et al. UK. 2022 (80) | Transition in/out of poverty | working-age adults aged 25–64 years (n=45 497) | Cohort | Marginal structural modelling | UK Household Longitudinal Study | Psychological distress; GHQ-12. Using cut-off | Negative effect of transitioning into poverty and positive effect of moving out of poverty on mental health. Effect of moving into poverty stronger. | Effect stronger in women and those with low education. | – | 22 |
| Swift et al, 2020; USA (82) | Change in Employment and income due to global financial crisis 2008 | Adults that had been consistently employed full-time prior to financial crisis (n=1307 (employment) and 1563 (income)) | Cohort | Regression using fixed effects. | Coronary Artery Risk Development in Young Adults (CARDIA) study; 2 waves: 2005-2010 | Depressive symptoms; CESD-20 | Both transitioning into unemployment and drop in income associated with higher depressive symptoms. | | | 24 |

*(Continued)*

Table 2. (Continued)

| Reference & country | Intervention | Target population, sample size | Study design | Analysis | Data source | Outcome and measure | Results | Sub-group analysis (moderation) | Mechanisms (mediation) | QAT |
|---|---|---|---|---|---|---|---|---|---|---|
| McCarthy et al. 2018. Canada and USA (California) (85) | Changes in income and material hardship | Adults employed in hair styling, food service, sex work (n=442) | Cohort | Linear regression (mixed effects) | Panel data – 2 waves 2003 – 2008. | Depression; Beck depression Inventory. Mental health – 2 items | Non-significant association between changes in income and depression or mental health | | Material hardship mediates effect of income on depression | 18 |
| Wickham et al. 2017. UK (79) | Transition into poverty (<60% of national median household income) | Mothers (n=6063) | Cohort | Regression - discrete time hazards survival models | UK Millennium Cohort Study. Included children at age 3 years; 5 years; 7 years; 11 years – 4 waves | Psychological distress; K-6 | Positive association with psychological distress. | – | Employment status was not a mediator | 20 |
| Curl and Kearns. 2015. UK (83) | Changes in financial difficulty in relation to austerity and Global Financial crisis | Residents of deprived neighbourhoods. Waves 1–2: n=840; Waves 2–3: n=1,021; Waves 1–3: n=936 | Repeated cross-sectional as well as retrospectively constructed longitudinal sample. | Regression models | Household survey conducted across 15 communities in Glasgow. All of the communities lie within the 15% most deprived in Scotland. 2006, 2008 and 2011 | Mental health component of SF-12 | Increased financial difficulty (affording fuel, food and council tax) significantly associated with decline in mental health. Decreased financial difficulty associated with improved mental health. | – | – | 17 |
| McInerney et a. 2013. US (84) | Global Financial crisis 2007-08. Changes in wealth post stock-market crash October 2008. | Adults aged ≥55 years born no later than 1953. n=20,040 | Cohort | Regression – first difference models | Data from Health and Retirement Study – 2006 and 2008 waves. | CES-D-20 and Anti-depressant prescriptions | The stock-market crash reduced wealth and increased depressive symptoms and use of antidepressants | Effects were largest among respondents with high levels of stock holdings prior to the crash. | – | 19 |
| Sareen et al, USA. 2011 (75) | change in household income | US adult population ≥18yrs (n=34,653) | Cohort | Multiple logistic regression | NESARC - nationally representative survey of the US population; 2001-02 and 2004-05 (2 waves) | Incident depression; DSM-IV | Decrease in income associated with higher risk of incident depression. No association with income increase | None observed – tested for age, sex | – | 23 |
| Prause et al. 2009. USA (74) | Income volatility | adults aged 30-40 yrs (n=4,493) | Cohort | regression (OLS) | National Longitudinal Survey of Youth | Depressive symptoms; CES-D | Decrease in income associated with higher depressive symptoms | No association when absolute volatility was low. Income level buffered effects. | | 23 |

(Continued)

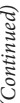

**Table 2.** (Continued)

| Reference & country | Intervention | Target population, sample size | Study design | Analysis | Data source | Outcome and measure | Results | Sub-group analysis (moderation) | Mechanisms (mediation) | QAT |
|---|---|---|---|---|---|---|---|---|---|---|
| Benzeval and Judge, 2001. UK (76) | Change in income level | Adults (n=5281) | Cohort | Multi-level models | British Household Panel Survey, 6 waves 1991–96/7 | Psychological distress; GHQ | Decreased income >30% associated with higher psychological distress. No association with >30% increase in income and health | | | 22 |
| **Employment** | | | | | | | | | | |
| Matsubayashi et al. 2022. Japan (98) | Adverse changes in employment: job loss or reduction in working hours during covid-19 pandemic | general population (n=9000) | Cross-sectional | Logistic regression | Online survey conducted between April 2020 and February 2021 | Depressive symptoms. PHQ-9 | Depressive symptoms were more prevalent among those who had recently experienced drastic changes in employment | No difference in effect between men and women. | – | |
| Jonsson et al. 2021. Sweden (102) | Low quality employment trajectories. | 16–65 years (n=2,743,764) | Cohort. Exposure established 2005-2009 and outcome in 2010-2017 | cox regression. | Register-study based on the Swedish Work, Illness, and Labour-market Participation cohort. 2005-2017 | Incidence of Depression based on healthcare registry data on ICD-10 codes (for depression (F32-F33)) | Low quality employment trajectories (approx. 21% of total population) associated with higher incidence of depression. | Women had slightly higher risks than men. | – | 23 |
| Hoven et al. 2021 France. (101) | Trajectories of employment: job discontinuity, job instability and cumulative disadvantage | Adults aged >45 years who were employed at baseline (n=13716 men and 12767 women) | cohort | Logistic regression | Annual CONSTANCES panel. Baseline in 2012 | Depression in the past 12 months. Single question. | participants who previously experienced unemployment periods, temporary employment or years out of paid work had higher risk of depression | Strength of effects different in men and women | Limited effect of psychosocial stress at work on this association. | 18 |

*(Continued)*

**Table 2.** (Continued)

| Reference & country | Intervention | Target population, sample size | Study design | Analysis | Data source | Outcome and measure | Results | Sub-group analysis (moderation) | Mechanisms (mediation) | QAT |
|---|---|---|---|---|---|---|---|---|---|---|
| Eisenberg-Guyot et al. 2020. USA (100) | Trajectories of employment quality. | Adults aged >18 years (n=2,779) | cohort | log-linear Poisson generalized estimating equations | 1985–2017 Panel Study of Income Dynamics | Mental health. Kessler-6 using cut-off | Poor employment quality associated with reduced mental health. Specifically: In men losing work was associated with reduced mental health. In women being 'minimally attached' to work (including low wages, part-time work etc) was associated with reduced mental health over time. | poor employment quality more common in ethnic minorities and lower educated. | | 19 |
| Brydsten et al. 2020. Sweden (103) | Precarious employment trajectories. | young people at age 20 years, living in Sweden in 1998, and followed their labour market transitions across 17 years until mid-life (n=98 634) | cohort | Logistic regression | Swedish longitudinal register data | Psychiatric admission based on ICD-10 codes F00-99) | A more precarious labour market trajectory was associated with higher risk of mental ill health in mid-life | Precarious labour market trajectory was more common among migrants, also in the second generation. | – | 23 |
| Farré et al. 2018. Spain (97) | Increased Unemployment rates in a context of economic crisis (Great Financial Crisis) | Working aged adults 16-65 years | Natural experiment Repeated cross-sectional data. | Instrumental variable (collapse of construction sector due to GFC); linear regression | Spanish National Health Survey 2006 and 2011 | Doctor diagnosed depression, chronic anxiety or other mental disorder. Psychological distress. GHQ-12 | Increase in unemployment rate was associated with higher psychological distress and higher probability of doctor diagnosed mental disorders. | – | – | 18 |
| Carlier et al. 2018. Netherlands (87) | Re-employment programme | Persons receiving social security benefits (n=869) | Quasi-experimental; propensity score matching | GEE | Participants assigned to either regular re-employment programme versus interdisciplinary programme | Anxiety and depression. K-10 | Symptoms of depression and anxiety decreased in those who gained employment | – | – | 18 |

(Continued)

| Reference & country | Intervention | Target population, sample size | Study design | Analysis | Data source | Outcome and measure | Results | Sub-group analysis (moderation) | Mechanisms (mediation) | QAT |
|---|---|---|---|---|---|---|---|---|---|---|
| Barbaglia et al, 2014. Netherlands (77) | Household income reduction & Job Loss | Adults 18-64 years (n=5303) | Cohort | Logistic regression models | Netherlands Mental Health Survey and Incidence Study-2 (NEMESIS-2), representative population-based study 2007–9/2010-12 | Incidence of mood disorder. Composite International Diagnostic Interview (CIDI) 3.0 | Household income loss and loss of employment associated with higher risk of mood disorder: major depression, dysthymia, bipolar disorder | Job loss increased risk among men only and household income reductions increased risk among women only | | 23 |
| McKenzie et al, New Zealand. 2013 (86) | Changes in Socio-economic status: labour force status; household income; area and individual deprivation | Population 15-60yrs at wave 3 (n=11 855) | Cohort | Fixed effects regression modelling | Survey of Family, Income & Employment; 2004-05, 2006-07 and 2008-09 (3 waves) | Mental Health (MHI-5) and psychological distress (K-10) | Becoming unemployed and increased individual deprivation associated with declined MH and increased psychological distress. Household income and area deprivation not associated with either outcome. | – | – | 21 |
| Kim et al. 2012. South Korea (99) | Change in employment status | People (aged >18 years) employed in permanent work or classified as 'precarious workers' (n=2891) | cohort | Logistic regression | Annual Korean Welfare Panel Study. Data from 2nd and 3rd wave (April–July of 2007 and 2008) | New-onset depressive symptoms. CES-D-11 | Workers who became unemployed following precarious employment had higher odds of developing depressive symptoms. | Main finding was stronger in women than men. In women only, becoming unemployed from precarious employment also had higher odds of depressive symptoms. | – | 22 |
| Zabkiewicz. 2010. USA (106) | Gaining employment | Single mothers receiving TANF benefits (n=419) | Cohort, 4yr follow-up | GEE | WCLS. | Depression in the past week based on cut-off. Brief Symptom Inventory. | Becoming employed was associated with decline of depression | Odds of depression was higher in women with ≥3 children and women with heavy drug use. | | 22 |
| Zabkiewicz and Schmidt. 2009; USA (107) | Gaining employment effect on women with drinking problem | Welfare mothers receiving Temporary Aid to Needy Families (TANF) (N=419). | Cohort, 4yr follow-up | GEE | Welfare Client Longitudinal Study California (WCLS). Yearly interview (2001 - for 4 years) | Depression in the past week based on cut-off. Brief Symptom Inventory. | Lower odds of depression with gaining full-time employment | No changes in women who were heavy drinkers at baseline. | – | 20 |

(Continued)

Table 2. (Continued)

| Reference & country | Intervention | Target population, sample size | Study design | Analysis | Data source | Outcome and measure | Results | Sub-group analysis (moderation) | Mechanisms (mediation) | QAT |
|---|---|---|---|---|---|---|---|---|---|---|
| Salm 2009. USA (111) | Job loss due to business closure | Older Adults (mean age 55 yrs) n=6867 | cohort | Difference-in-difference | Health and Retirement Study 1994 to 2002 (4 waves) | Depressive symptoms CES-D-8 | Job loss was not associated with depressive symptoms | – | – | 22 |
| Lorant et al, 2007. Belgium (78) | Change in Socio-economic status: employment status; material standard; social relationships; | Adult population (n= 11 909) | Cohort | Fixed effect model | Belgian House-holds Panel Survey; eight waves 1992–1999 | Depressive symptoms and depressed mood; Health and Daily Living Form. | Increased financial strain, deprivation or becoming poor associated with higher depression. Increased income or becoming unemployed were not associated with outcome. | Stronger effects on women; and on lower income category (for financial strain, becoming poor) | - | 23 |
| Gallo et al, 2000, USA (110) | Involuntary job loss in continually employed persons. | Workers aged 51 to 61 years at baseline. Continuously employed (n=2,907) or Involuntary job loss n=209) | cohort follow-up over 2 years | OLS regression | Health and Retirement Survey 1992-1994 (2 waves) | Depressive symptoms. CES-D-CES-D-8 | Involuntary job loss significantly associated with depressive symptoms. | | | 23 |
| Gallo et al, 2006. USA (109) | Involuntary job loss in continually employed persons. | Older workers who experienced job loss (n=231) compared to non-displaced individuals (n=3,324) | Cohort follow-up after 6 years | Instrumental variable (plant closing or layoff); linear regression | Health and Retirement Survey 1992-1998 (4 waves) | Depressive symptoms. CES-D-8 | No association in the total population at 6 year follow-up | Involuntary job loss only associated with depressive symptoms in those with low financial means. | – | 23 |
| Breslin and Mustard, 2003, Canada (108) | Employment transition | Working age population 18-55 yrs. n=6,096 | Cohort | Logistic regression | National Population survey Canada 1994-1996 | Composite International Diagnostic Interview (CIDI) - short form | Becoming unemployed was associated with depression over time. | Effect only observed in older population (31-55yrs) | | 23 |

(Continued)

**Table 2.** (Continued)

| Reference & country | Intervention | Target population, sample size | Study design | Analysis | Data source | Outcome and measure | Results | Sub-group analysis (moderation) | Mechanisms (mediation) | QAT |
|---|---|---|---|---|---|---|---|---|---|---|
| Pevalin and Goldberg, 2003, UK (91) | Change in employment status | Adults ≥16 years at baseline | Cohort with yearly follow-up | Logistic regression | British Household Panel Survey from 1991 to 1998. 8 waves | Psychological distress. GHQ-12 | Loss of job associated with new onset psychological distress. Lower chance of recovery in those that remained unemployed. Higher recovery in those that gained employment. | – | – | 22 |
| Strandh. 2000. Sweden (92) | The impact of different exit routes from unemployment | Persons unemployed at baseline. | cohort | OLS regression | 'Long-term Unemployment Project. 2 waves 1997 and 1998 | Mental health. GHQ-12 | Exiting unemployment status was associated with improved mental health. | Larger effect on mental health if entering permanent v temporary employment; starting university education vs lower educational levels. Exit to sick leave reduces mental well-being; exit to early retirement did no change mental health | – | 22 |
| Claussen et al 1993 Norway (89) | Gaining employment | Registered unemployed for more than 12 weeks (n=291) | Cohort 1988-1990 (2 years) | regression analysis | Random sample of registered unemployed compared to general population | GHQ and Hopkins symptom inventory | Lower depression symptoms and odds of depression with gaining employment – small effect | | | 19 |
| Claussen, 1999. Norway (88) | Gaining employment | Registered unemployed for more than 12 weeks (n=210) | Cohort follow-up 1993 (5 years) | Regression analysis | As above | GHQ and Hopkins symptom inventory | Recovery in mental health from 1988 to 1993 for the re-employed and worsening for the still unemployed. Effect larger than 1993 study | – | – | 19 |

*(Continued)*

| Reference & country | Intervention | Target population, sample size | Study design | Analysis | Data source | Outcome and measure | Results | Sub-group analysis (moderation) | Mechanisms (mediation) | QAT |
|---|---|---|---|---|---|---|---|---|---|---|
| Ali and Avison. 1997. Canada (104) | Employment transition | Mothers with at least one child <17 years. (n=948) | Cohort | Regression (linear) | Municipal data to identify single mothers and stratified sample to include representation of household income (area-level) | Depressive symptoms. CES-D-20. | Neither transitions in or out of employment were associated with changes in depressive symptoms for the combined sample. | In single mothers: transition out of paid work associated with increased depressive symptoms; stably unemployed versus those who enter paid work – no impact of gaining employment | Psychological resources (mastery; self-esteem) and financial strain partly explain result in single mothers transitioning out of work. | 22 |
| Morrell. 1994. Australia (93) | Employment transition | People aged 16-25 years. (n= 2403 and n= 8995) | cohort | Bayesian probabilistic approach was adopted. | Annual Australian Longitudinal Survey of registered unemployed young people 1984 -87. Plus annual survey of 16-25 year old general population 1985-88. | Psychological distress. GHQ-12. | Employed people who become unemployed had higher relative risk of psychological distress. Unemployed who became employed had lower relative risk of psychological distress | – | – | 20 |
| Graetz. 1993. Australia (94) | Employment transition | People aged 16-25 years (n=6151) | cohort | t-test, pairwise comparison | Australian Longitudinal Survey of general population. 1985 and followed up 3 years. | GHQ-12 | Employed people who become unemployed had significant increase in GHQ scores. Unemployed who became employed had decrease in GHQ scores | – | Quality of work | 21 |
| Lahelma. (1992) Finland (96) | Gaining employment | Registered job seekers in industry aged 25 to 49 years. | Cohort | Regression (logistic) | Panel survey | Psychological distress. GHQ-12 with cut-off 3-12 points | Reduced psychological distress for unemployed that became employed | Adverse impact unemployment stronger among men. | Low financial status had minimal impact | 19 |
| Dew et al. 1992. USA (105) | Involuntary job loss in women | Women aged 16-65 yrs employed >6months in a factory (n=145) | Cohort – follow-up over 1 year | Linear regression | Panel data | Depressive symptoms. Hopkins symptom checklist. | The occurrence and duration of being laid-off was associated with elevated depressive symptoms. | Among women who were laid-off financial hardship and lack of partner support were associated with higher symptoms. | – | 19 |

*(Continued)*

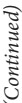

| Reference & country | Intervention | Target population, sample size | Study design | Analysis | Data source | Outcome and measure | Results | Sub-group analysis (moderation) | Mechanisms (mediation) | QAT |
|---|---|---|---|---|---|---|---|---|---|---|
| Kessler et al. 1989, USA (90) | Gaining Employment | Adult population in Michigan. (n=414) | Cohort 1984-85 (1 year) | Regression (linear and logistic) | Stratified sample: currently unemployed; previously unemployed; stably employed. | Depression; sub-scale of the SCL-90. Using established cut-off. | Lower probability of depression in the re-employed vs the still unemployed at follow-up | – | – | 17 |
| Frese and Mohr. 1987. Germany (95) | Changes in employment status | Unemployed blue-collar workers aged>45 (n=51) | Cohort | Ancova | Panel data. 1975 and 1977 | Depressive symptoms. Hamburg depressive scale. | Higher depressive symptoms in those with repeated unemployment (i.e. got work and lost it again); lower depressive symptoms in those gaining employment or becoming retired. | – | | 13 |
| **Job security** | | | | | | | | | | |
| Hanson et al. 2015; Sweden (113) | employment security | Persons 'gainfully employed' at the baseline study in 2003 aged 16-64 years. n=6275 | Cohort | GEE | Swedish Longitudinal Occupational Survey of Health (SLOSH) study - 2003 or 2005 and FU in 2008, 2010 and 2012 | Depression. subscale from the (Hopkins) Symptom Checklist - the SCL-CD6 | Increased odds of depression with threat of job loss and repeated threat of job loss | – | – | 23 |
| Ferrie et al, 2002; UK (112) | Employment security | Office staff, aged 35–55, working in civil service (n=3,360). | Cohort | Regression (linear and logistic) | Whitehall II phase 4 (1995/6) & 5 (1997/9) | Depression. GHQ-12 – sub-scale depression. | In men relative to securely employed: highest level of depression in those with chronically insecure job. Moving from insecure to secure job had lower association with depression, effect size similar to moving from secure to insecure job. | Findings among women similar but statistical significance impacted by smaller number | | 21 |

(Continued)

Table 2. (Continued)

| Reference & country | Intervention | Target population, sample size | Study design | Analysis | Data source | Outcome and measure | Results | Sub-group analysis (moderation) | Mechanisms (mediation) | QAT |
|---|---|---|---|---|---|---|---|---|---|---|
| **Housing individual-level: adequacy** | | | | | | | | | | |
| Petersen et al. 2022. UK (115) | Selective Licencing aimed to improve quality of rental housing. Introduced 2012-2018 | 33 local authorities in London. | Area-level data 2011-2018. Ecological-level data | Difference-in-difference with propensity matched areas. | National Health Service data and Department for Work and Pensions data. | Small Area Mental Health Index scores by year and small area. Based on mental healthcare from multiple sources. | Improvements in area-based mental health outcomes in areas with Selective Licencing: reduced Antidepressant treatment days per population; reduced mental health benefit receipt and reduced proportion with depression | – | – | 20 |
| Pevalin et al. 2017. UK (116) | The cumulative impact of housing problems (light; adequate heating; condensation; leaky roof; damp walls; and rot) | Adults (n=8,365) | Cohort | Pooled OLS regression and lagged-difference model. | British Household Panel Survey. 13 annual waves (1996 to 2008) | Psychological distress. GHQ-12 | Past housing problems was associated with current poorer mental health. Living in persistently poor housing associated with poor mental health over and above the effect of current housing problems. | Mental health effects of persistent housing problems harm social renters and outright owners the most. | – | 21 |
| Blackman et al. 2003, UK (114) | Medical priority rehousing (MPR) | Adults with health care problems requiring rehousing (n=227) | Quasi-experiment – Cohort (9-12 month f.u.) | 2x2 table comparison; McNemar's test | All new applicants to the council's MPR waiting list (Newcastle upon Tyne) | Mental health; depression. SF-36 | Prevalence of mental health problems and odds of depression among the rehoused group decreased | – | – | 15 |
| **Housing individual-level: affordability and stability** | | | | | | | | | | |
| Denary, et al. 2021 USA. (117) | Access to rental assistance | Low-income adults. Homeless; resided in a low-income census tract; received housing or food assistance; received medicaid (n=84) | Cohort 2017 f.u. bi-annual for 2 yrs. | Regression (fixed effects, logistic) | Justice, Housing, and Health Study (JustHouHS) | Psychological distress. K-10 cut-off ≥30 | No significant decrease in psychological distress with obtaining rental assistance. | – | Homelessness may be an important pathway through which rental assistance reduces psychological distress | 18 |

*(Continued)*

**Table 2.** (Continued)

| Reference & country | Intervention | Target population, sample size | Study design | Analysis | Data source | Outcome and measure | Results | Sub-group analysis (moderation) | Mechanisms (mediation) | QAT |
|---|---|---|---|---|---|---|---|---|---|---|
| Kim et al. 2021. South Korea (125) | Impact of changes in housing tenure status and affordability of housing. | Adults aged >20 years who could afford housing. (n= 9956 respondents in 5507 households) | Cohort. Data from wave 10 (2015) to wave 15 (2020) | generalized estimating equation (GEE) model, | Annual Korea Welfare Panel Study. | Depressive symptoms. CES-D-11 | Transitioning from home ownership to renting or if housing became unaffordable were both associated with increased depressive symptoms. Depressive symptoms reduced in new homeowners and those who no longer had unaffordable housing. | Results differed according to age category. | – | 21 |
| Hoke and Boen. 2021. USA (120) | Housing eviction | Young adults aged on average 24yrs, n=9029 | Cohort; propensity score matching | Regression – Generalised linear models, | National Longitudinal Study of Adolescent to Adult Health (1994–2009) (3 waves) | Depressive symptoms. CES-D-9 | Experiencing eviction associated with increase in depressive symptoms. | – | Perceived social stress mediated partly association | 23 |
| Hatch and Yun. 2021. USA (119) | Housing eviction | Adolescents (n=11513) | Cohort | Logistic regression | National Longitudinal Study of Adolescent to Adult Health. 1994–95, 2001 and 2008. | Doctor diagnosed depression. Single question | Having experienced housing eviction (ever) associated with depression. additionally, the effect was stronger in those experiencing more recent eviction. | Stronger negative effect on women and on white versus non-white participants. | – | 19 |
| Leifheit et al. 2021. USA (121) | Eviction moratorium in context of covid-19. | Persons living in rented housing (n=8277) | Cohort. Used variation in state-level eviction moratorium. | linear regression with individual fixed effects | Understanding Coronavirus in America Survey – online panel survey in 2020 using data pre- federal moratorium | Depressive symptoms. PHQ-4 | Prevalence of mental distress was 17.7%. In states with weak moratoriums 17.0% and in states with strong moratoriums 15.5% | strong moratoriums associated with larger reductions in mental distress in Hispanic and non-Hispanic White than in black participants. | – | 22 |

*(Continued)*

**Table 2.** (Continued)

| Reference & country | Intervention | Target population, sample size | Study design | Analysis | Data source | Outcome and measure | Results | Sub-group analysis (moderation) | Mechanisms (mediation) | QAT |
|---|---|---|---|---|---|---|---|---|---|---|
| Desmond and Kimbro. 2015. USA (118) | Housing eviction | Low income Mothers who were renting at baseline (n= 2,676) | Cohort. Propensity score matching. | Logistic regression | Fragile Families and Child Wellbeing Study (FFCWS), a survey that follows a birth cohort of new parents and their children. (Wave I–1998–2000; wave 2 year one; wave 3 year 3; wave 4 at year 5. | Depression. Composite International Diagnostic Interview Short Form | Mothers who were evicted the previous year were more likely to have depression. Evidence that at least two years after eviction mothers still experienced significantly higher rates of depression than their peers | – | – | 20 |
| Fowler et al. 2015. USA (126) | Housing instability (number of different addresses) and adolescent health | Adolescents (n=5596 at baseline) | Cohort | Multi-level regression | National Longitudinal Study of Adolescent Health. 3 waves 1994–95; 2001–02 and 2008–09 | Depressive symptoms. CES-D-9 | Number of housing moves predicted odds of depression. Every move increased the odds of depression by 10% | – | Effect appears independent of family structure/ instability. | 21 |
| Alley et al. 2011. USA (124) | Mortgage delinquency and health in context of housing crisis. | Adults aged ≥50 years (n=2239) | Cohort | Logistic regression | Health and Retirement study. 2006 and 2008 | Depressive symptoms. CES-D-8 with cut-off. | Those who fell behind with mortgage payments had higher odds of developing incident depressive symptoms | – | – | 20 |
| Suglia at al. 2011. USA (127) | Housing instability (moving ≥2 times in 2 years) | Women (n=2,104) | Cohort | Logistic regression | Fragile Families and Child Wellbeing Study | Composite International Diagnostic Interview-Short Form | Experiencing housing instability associated with higher odds of depression at follow-up. | – | – | 20 |
| Pevalin. 2009. UK (122) | Housing eviction | Adults n=12 390 | Cohort; | Multivariate fixed effects regression models | British Household Panel Annual Survey – 17 waves starting in 1991 | Psychological distress. GHQ-12 | Housing repossession associated with increased risk of psychological distress; eviction from rented property no increased risk | – | – | 22 |

*(Continued)*

**Table 2.** (Continued)

| Reference & country | Intervention | Target population, sample size | Study design | Analysis | Data source | Outcome and measure | Results | Sub-group analysis (moderation) | Mechanisms (mediation) | QAT |
|---|---|---|---|---|---|---|---|---|---|---|
| Taylor, Pevalin and Todd. 2007. UK (123) | Unsustainable housing commitments | Adults (head of household) n=8185 | Cohort | Multivariate fixed effects regression models | British Household Panel Survey 1991–2003 | Psychological distress. GHQ-12 | Housing payment problems (entering arrears in mortgage or rental payment) associated with psychological distress | Similar effects for male or female heads of household. | – | 22 |
| **Housing collective-level: Economic disadvantage** | | | | | | | | | | |
| Leventhal and Brooks-Gunn, 2003. USA (128) | Moving To Opportunity experiment | Residents with children living in areas with poverty rates >40%, based on 1990 US Census. | Randomised intervention. 3yr f.u. | ITT (regression); Instrumental variable (Voucher to move) - TOT (2SLS) | follow-up evaluation of the New York City MTO site (n=794) | Parents' depressive mood - Depressive Mood Inventory and Hopkins Symptom Checklist | Compared to in-place controls: Parents (and children) who moved were less likely than to report depressive symptoms. | Program effects most pronounced for boys and for children aged 8 to 13 years. | – | 21 |
| **Housing collective-level: Urban regeneration** | | | | | | | | | | |
| Rose et al. 2023. UK (132) | Community Wealth Building programme | Residents of Preston - within the 20% most deprived local authorities in England | Natural experiment. Repeated cross-sectional | Difference-in-difference. | National statistics data comparing intervention versus similar areas - (2011–15) and post introduction of the programme (2016–19) | antidepressant prescribing, prevalence of depression | The programme was associated with reductions in the prescribing of antidepressants and prevalence of depression relative to the control areas. | – | – | 23 |
| Greene, et al 2020. UK (136) | 'Communities First' area-wide regeneration programme – examination of mediation by neighbourhood social factors. | Residents (≥18yrs) of Caerphilly County Borough (n=8,394) | Natural experiment – Cohort 2001-2008 | Propensity matching. OLS Regression | linked data: Caerphilly County Borough Council; the Secure Anonymised Information Linkage Databank; The eCATALyST study (prospective Cohort study) | Mental health. SF-36 MHI-5 scale. | See White et al 2017 | – | Improvement was explained by improvements in neighbourhood belonging, quality and reductions in disorder. Small effect of social cohesion | 20 |
| Timmermans et al. 2020. Netherlands (140) | Urban regeneration programme "Dutch District Approach" | Adults aged >55 years, n=1092 | Cohort | Difference-in-Difference with linear regression models and linear probability models (dichotomous variables) | The Longitudinal Aging Study Amsterdam; waves 2005/06 and 2011/12 | Depressive symptoms; CES-D-20 | No evidence of effect in residents of regeneration areas compared to similarly deprived non-target areas. | – | – | 21 |

*(Continued)*

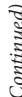

**Table 2.** (Continued)

| Reference & country | Intervention | Target population, sample size | Study design | Analysis | Data source | Outcome and measure | Results | Sub-group analysis (moderation) | Mechanisms (mediation) | QAT |
|---|---|---|---|---|---|---|---|---|---|---|
| White et al. 2017. UK (135) | 'Communities First' area-wide regeneration programme | Residents (≥18yrs) of Caerphilly County Borough (n=8,394) | Natural experiment – Cohort 2001-2008 | Propensity matching. OLS Regression | linked data: Caerphilly County Borough Council; the Secure Anonymised Information Linkage Databank; The eCATALyST study (prospective Cohort study) | Mental health. SF-36 MHI-5 scale. | Regeneration associated with an improvement in the mental health. Evidence of a dose-response association with length of residence. | – | – | 20 |
| Egan et al 2016. UK (138) | Urban renewal - level of renewal investment | Residents of deprived neighbourhoods n-1006 | Cohort | Difference-in-Difference. Multiple regression Accounting for area-level dependence | Construced Cohort from GoWell repeated cross-sectional study 2006-2011 | SF-12 mental health sub-component | Improvement in mental health in the higher investment areas. | No differences based on education or country of birth. | | 20 |
| Walthery et al. 2015; UK (134) | Area regeneration programme (NDC) | Residents in deprived areas aged ≥16 years (n=11,648) | Repeated cross-sectional; matching at area-level - 2002, 2004, 2006 and 2008 | Latent growth models | household panel survey on NDC areas and non-contiguous disadvantaged comparator areas within the same local authority | Mental Health Inventory (MHI-5) | No evidence of effect in NDC compared to comparator non-NDC communities. | Respondents in poorer socio economic circumstances in NDC areas fared better than those in comparator areas. | – | 21 |
| Jongeneel-Grimen et al. 2015. Netherlands (139) | Urban Regeneration programme "Dutch District Approach" | Adults ≥18 years (n=46 240) | Repeated crosssectional study. Pre-intervention (2004-June 2008) and intervention (July 2008– 2011) | Multilevel logistic regression. Propensity score matching of residential areas. | Annual Health Interview Survey, for the period 2004–2011 | Mental Health Inventory (MHI-5) | No evidence of effect in regeneration areas compared to similarly deprived non-target areas. | Some effect in women. More intensive programme showed improvement while comparable control districts experienced a deterioration. | – | 22 |
| Stafford et al. 2014; UK (133) | Area regeneration programme (New Deal for Communities NDC) | Adults (n=28,984-29,615) | Repeated crosssectional; matching at area-level in 2002, 2004, 2006 and 2008 | Multi-level regression | Representative Household surveys (HSE) and household survey data collected by the NDC team in intervention and deprived comparator areas. | Mental health. MHI-5 in the NDC intervention surveys. GHQ-12 in the HSE. | No evidence of effect NDC compared to general population stratified (based on deprivation level) or when compared to comparator high deprivation non-NDC communities. | – | – | 19 |

*(Continued)*

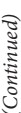

| Reference & country | Intervention | Target population, sample size | Study design | Analysis | Data source | Outcome and measure | Results | Sub-group analysis (moderation) | Mechanisms (mediation) | QAT |
|---|---|---|---|---|---|---|---|---|---|---|
| Egan et al 2013. UK (137) | Urban renewal - housing demolition or improvement | Residents of deprived neighbourhoods n=1041 | Cohort | Difference-in-Difference. multiple regression models accounting for clustering based on area. | Constructed Cohort from GoWell repeated cross-sectional survey 2006–2008 | SF-12 mental health sub-component | Small improvements in mental health with housing improvement. | – | – | 20 |
| Huxley et al. 2004. UK (131) | Urban regeneration programme (Single Regeneration Budget SRB) | Adults. (n=1344) | Cohort; matching at area-level 1999-2001 | Analysis of covariance | Random sample from electoral register. | Psychological distress, GHQ-12 | No evidence of effect in SRB areas versus comparative deprived neighbourhoods. | – | – | 22 |
| Harvey. 2001, UK (130) | Neighbourhood and housing renovation | Adults >16 years old. (n=166 at both time points) | Cohort pre-renewal (1992) and post-renewals 5 years later | Multi-variate regression. | All residents of renewal area sampled. | Psychological distress measured using 7 questionnaire items | Drop in number of participants reporting mental health problems at follow-up. | – | – | 15 |
| **Collective Material Insecurity: Financial Crisis** | | | | | | | | | | |
| Villarroel et al. 2022. Ireland (155) | Global financial crisis 2007–08 | Fathers of children receiving child benefit. (n=6251) | Cohort. | Binary logistic regression. | Growing up in Ireland, longitudinal study. 2008– 2013 | Depression. CES-D-8 using cut-off | Increased depression post-recession. | Effect seen only in Irish fathers. No effect in non-Irish and African-origin fathers | – | 22 |
| Cherrie et al. 2021. UK (154) | Global Financial crisis 2007–08 – area-level employment | Working aged persons aged 16-60 years (n=86500) | Cohort. 2009–15 | Multi-level logistic regression | Scottish Longitudinal Study | New cases depression. Anti-depressant prescription -Linkage data | Increased odds of beginning a new course of antidepressants if living in areas with declined Full-time employment | | population income loss due to welfare reforms explained 50% of the effect | 26 |
| Nour et al. 2017. Canada (153) | Global Financial crisis 2007–08. Exposure to austerity policies. | Working age population aged 15-64 years. (over-all n=306 623) | Repeated cross-sectional - 7 cycles (2007–2013) | Logistic regression | Canadian Community Health Survey | Mood disorders. Reporting a health professional diagnosis of MD | GFC exposure not associated with depression initially; subsequent austerity period was associated with higher depression. | interaction terms for sex, age and income quintile – no indication of differences in population subgroups | income adequacy, employment status and home ownership did not mediate association | 20 |

*(Continued)*

**Table 2.** (Continued)

| Reference & country | Intervention | Target population, sample size | Study design | Analysis | Data source | Outcome and measure | Results | Sub-group analysis (moderation) | Mechanisms (mediation) | QAT |
|---|---|---|---|---|---|---|---|---|---|---|
| Drydakis. 2015. Greece (149) | Global Financial Crisis 2007-08 influence on the role of unemployment in mental health. | Adults aged 16-65 years (n=1418-1447 men and 1553-1510 women) | cohort | Fixed effect linear models | Longitudinal Labor Market Study. Yearly survey since 2008. | Depressive symptoms. CES-D-20 | During the 2010-13 period, unemployment led to higher mental health deterioration compared to the 2008-09 period. | Effect of unemployment was larger in women than men. | – | 21 |
| Modrek et al. 2015; USA (147) | Global Financial crisis 2007-08. | Workers from a multi-site (25 sites) US manufacturing firm who were continuously employed during study period. n= 11625 and n=10242 | Cohort | Regression | 2 Cohorts: employee panels: Jan 2007-Dec 2010 and Jan 2007-Dec 2012 | Depression. ICD-9-CM codes to identify encounters with any mental health component. pharmacy claims data to calculate use of medications. | Mental health inpatient and outpatient visits and antidepressant use increased after 2009 in total population. | Use of antidepressants was higher in high layoff plants than in other plants (ie in context of job insecurity). | – | 21 |
| Buffel et al. 2015. 20 European countries (148) | Global Financial Crisis 2007-08. Changes in country-level unemployment rates and change in GDP | Working-age population (20–65 years) n=51,679 | Repeated cross-sectional. | multi-level models considering the individual nested in country years and in countries | European Social Survey's Round 3 (2006) and Round 6 (2012), | Depressive Symptoms: CES-D-8 | In countries with a high increase in unemployment rate a higher likelihood of being depressed was observed. In countries with a high GDP respondents were less likely to be depressed than in countries with lower GDP. | Stronger effect in men and for ages 35–49 years. | – | 21 |
| Aguilar-Palacio et al. 2015. Spain (141) | Global Financial Crisis 2007-08 in relation to unemployment | Young adults aged 16-24 years. (n=2168 in 2006; 1533 in 2011-12) | Repeated cross-sectional | Logistic regression. | Spanish National Health Surveys 2006 and 2011-12 | Mental health. GHQ-12. | Association between mental health and unemployment did not differ in 2012 compared to 2006 in men. Unemployed women had better mental health in 2012. | Differences between men and women. | – | 18 |

*(Continued)*

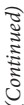

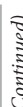

| Reference & country | Intervention | Target population, sample size | Study design | Analysis | Data source | Outcome and measure | Results | Sub-group analysis (moderation) | Mechanisms (mediation) | QAT |
|---|---|---|---|---|---|---|---|---|---|---|
| Gotsens et al. 2015. Spain (156) | Global Financial Crisis 2007-08. Change in ethnic health inequalities | Adults aged 16–64 years (n= 23,760 and 16,616) | Repeated cross-sectional | Poisson regression. | Spanish National Health Survey. 2006-07 and 2011-12 | Mental health. GHQ-12 | Mental health status of immigrants that arrived before 2006 evolved less favourably than that of natives. | Difference only significant in women. | – | 20 |
| Malard et al. 2015. France (143) | Global Financial crisis 2007-08. Effect on working population. | Adults working in both time periods aged 20-74 years. (n=2679 men; 2921 women) | Cohort. | Generalized estimating equations for logistic regression | National representative Santé et Itinéraire Professionnel survey | Major depressive episode. Mini International Neuropsychiatric Interview | No changes in major depressive episode between survey periods. | No differences based on age, ethnic origin, occupation, public/private sector, self-employed/ employee status and work contract. Self-employed women had a lower prevalence of MDE in 2010 compared to 2006 | – | 22 |
| Barr et al. 2015. UK (152) | Global Financial crisis and austerity measures; trends in unemployment and wages and trends in mental health. | Adults aged 18-59 yrs who participated in each of the quarterly surveys between 2004 and 2013. (n= 2,171,741) | Cohort | Linear regression | Quarterly Labour force Survey. | Mental health problems based on self- reported depression, bad nerves, anxiety, mental illness, phobias, panics or other nervous disorders. | Significant break in the trend (increase) in poor mental health between the last quarter of 2008 and the first quarter of 2009. | Widening gap between low and high educated post 2008. | Increase in area-level unemployment and decrease in area-level income explained 36% of decline in mental health. | 23 |
| Economou et al. 2013. Greece (146) | Global Financial crisis 2007-08. | adults aged 18-69 years n=2.197 and 2.256 | Repeated cross-sectional; weighting | 2x2 tables; chi-square test | Existing data from the 2008 nationwide cross-sectional study and in 2011 a random sample based on telephone listings was used. | Major depressive episode (MDE). Structured Clinical Interview (SCID-I) was used | Odds of MDE Depression higher in 2011 than in 2008: | More depressive symptoms in people who experience high economic distress | – | 20 |
| Astell-Burt et al.2013. UK (142) | Global Financial crisis 2007-8 in relation to rise in unemployment rates | Working age, women 16-59yrs; men 16-64yrs (n=1.36 million) | Repeated cross-sectional | Logit regression | UK quarterly Labour Force Survey. Pooled data from Jan 2006-Dec 2010 (20 quarters of survey) | Depression. single question | Reported depression did not change post GFC, despite increases in unemployment rates. | No differences on basis of geographical region, socio-economic inequalities were not exacerbated. | – | 23 |

*(Continued)*

**Table 2.** (Continued)

| Reference & country | Intervention | Target population, sample size | Study design | Analysis | Data source | Outcome and measure | Results | Sub-group analysis (moderation) | Mechanisms (mediation) | QAT |
|---|---|---|---|---|---|---|---|---|---|---|
| Bartoll et al. 2013. Spain (150) | Global financial crisis 2007-08 | Residents aged 16-64 years. (n=23760 in 2006–07; n=16616 in 2011–12) | Repeated cross-sectional. | Poisson regression | Spanish National Health Survey. 2006-07 and 2011-12 | Mental health. GHQ-12 based on cut-off | Increased prevalence poor mental health in men and slight decrease among women. Inequalities in mental health, based on social class, increased in men but not women. | Differences between men and women. | Employment status. | 20 |
| Gili et al. 2012. Spain (151) | Global Financial crisis 2007-08 | Persons visiting their General Practitioner | Repeated cross-sectional | Multivariate probability regression models | Data from randomly selected GP practices in Spain. 2006 and 2010. | Primary Care Evaluation of Mental Disorders instrument. | Higher rates of depression in 2020 (19.4 percentage point increase) | – | about 1/3 of the overall risk of depression could be attributed to the combined risks of individual and family unemployment, and mortgage payment difficulties | |
| Lee et al. 2010. Hong Kong (144) | Global Financial crisis 2007-08. | Adults 18-65 yrs. n=3016 and 2011 | Repeated cross-sectional. 2007 and 2009 | 2x2 tables; chi-square test, regression analysis | 2 general population surveys (telephone survey based on randomised selection of listings) | Major depressive episode (MDE). Diagnostic and Statistical Manual of Mental Disorders — 4th edition (DSM-IV) | Prevalence of MDE was higher in 2009 than in 2007. | Highest prevalence of MDE in those with higher loss of financial investments. | – | 21 |
| Madianos et al. 2008. Greece (145) | Global Financial crisis 2007-08. Personal Economic Hardship | adults aged 18-69 years (n=2,197 and 2,192) | Repeated cross-sectional. 2008 and 2009 | 2x2 tables; chi-square test | nationwide cross-sectional telephone surveys | Major depressive episode (MDE) in past month. The SCID I (semi-structured interview) | Higher prevalence of MDE in 2009. | MDE higher with personal economic hardship. | – | 20 |
| Rahmqvist and Carstensen. 1998. Sweden (157) | Economic Recession 1991. | Adults aged 20-34 years. (n=656) | Repeated cross-sectional. (1989, 1991, 1993, 1995) | Logistic regression | Population (2-yearly) surveys in the Swedish county of Östergötland. | Psychological distress based on 4 items. | Higher prevalence of psychological distress post 1991 with increasing prevalence up till 1995. | Higher prevalence in unemployed but increased trend also in employed. Odds were higher in men than women (1995) | – | 19 |

**Table 3.  Studies addressing social resources.**

| Reference & country | Aim/Intervention | Target population, sample size | Study design | Analysis | Data source | Outcome and measure | Results | Sub-group analysis (moderation) | Mechanisms (mediation) | QAT |
|---|---|---|---|---|---|---|---|---|---|---|
| **Social disorder** | | | | | | | | | | |
| Cooper et al. 2014. USA (129) | Housing relocation- local economic disadvantage (household income, poverty, education) and social disorder (alcohol outlet density and crime) | African American adults, living in a public housing complex ≥1yr (n = 172) | Cohort | Multi-level regression | Relocation of residents of 7 public housing complexes scheduled for demolition in 2008–2010. Baseline = pre-demolition and every 9 months after (4 waves). | Depressive symptoms. CES-D-20 | Moving to areas with lower area-level economic disadvantage and decline in perceived community violence associated with decline in depressive symptoms. | No differences between men and women | perceived community violence may mediate the relationship between economic conditions and depressive symptoms. | 20 |
| Mair et al. 2015. USA (159) | Change in neighbourhood-level social cohesion, safety, violence, stress, aesthetic environments | adults aged 45–85 at baseline (n = 548 MESA and n = 103 independent sample) | Cohort | Multi-level regression | MESA study and independent sample from phone survey. Survey 1 (Mesa) 2002–4 and 2005–7 Survey 2 (phone) 2006–2008 | Depressive symptoms CES-D-20 | Decreased depressive symptoms with increase in neighbourhood safety, aesthetic quality and increased social cohesion; Increased depressive symptoms with increased neighbourhood violence and stress. | – | – | 21 |
| Astell-Burt et al. 2015 Australia (160) | Neighbourhood-level crime | Adults 45 years and older. (n = 60, 404) | Cohort | Fixed effect regression models | 45 and Up Study (2006–2008). Follow-up: The Social Economic and Environmental Factors Study (2009–2010) | Psychological distress (K-10) | Increase risk of experiencing psychological distress with increased in the level of neighbourhood crime | Effect sizes were stronger in women. | | 23 |
| Dustmann and Fasani. 2014. UK (161) | Neighbourhood-level crime | Adults (n = 9400 and 16 000) | Cohort | Difference-in-Difference, regression. | British Household Panel Survey (BHPS) and the English Longitudinal Study of Ageing (ELSA). | Anxiety and depression. GHQ-12 (BHPS) and Psychosocial Health Module (ELSA) | increase in the overall local crime rate associated with increase in mental distress | Effect is stronger in women. | – | 23 |

*(Continued)*

**Social capital**

| Reference & country | Aim/ Intervention | Target population, sample size | Study design | Analysis | Data source | Outcome and measure | Results | Sub-group analysis (moderation) | Mechanisms (mediation) | QAT |
|---|---|---|---|---|---|---|---|---|---|---|
| Cruwys et al. 2014. Australia (162) | Social contact – community recreation group | People with a high risk for depression (n=52) | Non-randomised intervention [NB pre-post-design, no control group] | Regression analysis | participants in a community recreation group | Depression. Depression Anxiety Stress Scales (DASS-21) – depression sub-scale | Decline in depressive symptoms between Time 1 and Time 2, | | Social identification with the group predicted a more pronounced decline in depression symptoms | 13 |
| Murayama et al. 2015. Japan (164) | Intergenerational reading programme | Older adults (n=80) | Non-randomised intervention | Two-way ANOVA; | Research of Productivity by Intergenerational Sympathy (REPRINTS) program | Depressive symptoms. Geriatric Depression Scale-Short Version-Japanese -15 items | Participation in the program positively associated with sense of meaningfulness; which was associated with lower depressive mood. | – | – | 15 |
| Watanabe et al. 2019, Japan (165) | Change in Municipal-level health-related Social Capital | Functionally independent residents aged ≥65 years n=72,718 and 84,211 | Repeated cross-sectional. 2010–12 (2010) and 2016 | Linear regression | The Japan Gerontological Evaluation Study | Depressive symptoms. Geriatric Depression Scale-15 items | Inverse association depression with 10/14 indicators of Social Capital. | | | 21 |
| Lindstrom and Giordano. 2016 UK (81) | Change in **social capital** (trust and participation) in context of Global Financial crisis 2007–2008. And change in **financial status** | Adults (n=11,743) | Cohort. Waves 17 (2007) and 18 (2008) | Multilevel, longitudinal logistic regression | British Household Panel Survey | Psychological well-being. GHQ-12 | Lack of trust associated with deterioration in psychological wellbeing. No effect of social participation. Financial status: improved had lower risk, worsened had higher risk of psychological well-being. | No difference of effect of social capital based on financial strain. | | 21 |
| McGale et al. 2011, Ireland (163) | Team-based exercise intervention plus group with additional **social support** | Men aged 18–40 years (n=84) | Intervention study. Randomised 3 conditions: exercise; exercise plus social support; access to exercise facilities. | Mixed effects regression analysis | Participants in intervention study | Depressive symptoms: Beck Depression Inventory-II | Reduction of depressive symptoms in both exercise conditions. No difference in effect with additional social support. | | | 16 |

**Table 4. Summary of Findings.**

| | | Reduction in depression risk | Increase in depression risk | No association |
|---|---|---|---|---|
| **Societal arrangements** | | | | |
| Welfare reforms | Tax credit policies | 32, 31 | | 34, 33 |
| | Expanding access health insurance | 39, 40, 41 | | |
| | Work incentives and employment | 42, 43, 45, 47 | 44, 46, 47, 48, 49, 50, 51, 52 | |
| | Increased Pension age | | 63, 62 | |
| | Intro of old age pension | | | 64 |
| | Housing benefit sanctions | | 65 | |
| | Childcare benefits | | | 66 |
| | Parental leave | 61, 53, 55, 54, 56, 57, 58, 59 | | 53, 60 |
| Minimum wage | | 27 | | 36, 37, 38 |
| Educational policies | | 68 | | |
| Public transport policies | | 67 | | |
| Environmental interventions | noise exposure reduced | | | 70 |
| | Noise exposure increased | | 69 | |
| **Material resources** | | | | |
| Income | Income supplement | | | 71, 72, 73 |
| | reduction in income/ transition into poverty | | 74, 75, 76, 77, 78, 79, 80, 82, 83, 84, 86 153 | |
| | increase in income/out of poverty | 83 | | 75, 76 |
| | Income volatility (either direction) | | | 85 |
| Employment | Gaining employment | 87, 88, 89, 90, 93, 94, 95, 96, 106, | | 104 |
| | Losing employment | | 77, 86, 91, 93, 94, 95, 97, 99, 100, 104, 105, 108, 110, 152 | 111, 78, 109 |
| | Job insecurity/poor employment trajectories | | 98, 100, 101, 102, 103, 113, 112, | 104 |
| Housing quality: individual-level | Quality improved | 114, 115 | | |
| | Quality reduced | | 116 | |
| | Affordability | | 124 | 117 |
| | Instability of housing, or eviction | | 125, 120, 119, 121, 118, 126, 127, 122, 123 | |
| Housing conditions: collective-level | Economic disadvantage (improved) | 128, 132, 130 | | 87 |
| | Combination (urban regeneration) | 132, 135, 138, 137, 130 | | 140, 134, 131, 133, 139 |
| Financial/Economic Crisis | | | 152, 154, 155, 153, 149, 147, 148, 156, 146, 150, 151, 144, 145, 157 | 141, 143, 142, 150 |
| **Social Resources** | | | | |
| Social disorder (deterioration) | | | 159, 160, 161 | |
| Social disorder (improvement) | | 129, 159 | | |
| Social capital improved | | 136, 162, 164, 165 | | 163 |
| Social capital reduction | | | 81 | |

Numbers used in the table correspond to numbering of papers in tables 1 to 3, and list of references.

as a refundable tax credit to offset the rise in payroll, increasing household income. A study by Boyd-Swan et al. [31] among mothers suggests that expanding eligibility in the EITC in 1990 reduced depressive symptoms but only in married mothers. Shields-Zeeman et al. [32] leveraged variation in the size of the EITC to study the impact of changes in income and found a 2–3% of a standard deviation decrease in psychological distress per US$1000 increase in income. Two other studies found no effects of EITC payments and other income policies (including the Temporary Assistance to Needy Families (TANF) and Minimum wage policies) on psychological distress or depression risk in the short term [33,34].

**Minimum wage policies.** A UK study by Reeves et al [35] evaluated the effect of introduction of the National Minimum Wage legislation in 1999 and found lower probability of psychological distress among recipients compared to similar persons who were not covered under the conditions of the new legislation. Another UK study by Kronenberg et al [36] of the same policy and using the same data source however did not observe an association. The difference in findings between the two studies maybe due to difference in construction of treatment and control groups as well as in the way wages were measured. A third UK study, by Maxwell et al [37] studied increase in minimum wage phased in over three years (2016, 2017 and 2018) and found no association with mental health. In a US study, Buszkiewicz et al [38] used state-level variation in minimum wage legislation to evaluate the association with psychological distress but found no association in the population as a whole, or on the basis of gender, age, race/ethnicity (non-white/Hispanic or white), employment status.

**Expanding access to health insurance.** In 2008, the US state of Oregon opened a waiting list for a limited number of spots in its Medicaid program for low-income adults. This took the form of a lottery among the people who signed up. Finkelstein et al. [39] showed that in the first year after the lottery, winners had 10% higher probability of screening negative for depressive symptoms relative to the control group. More recently, in the context of the covid-19 pandemic, a US study by Mukhopadhyay compared data from states that expanded Medicaid entitlements with states that did not introduce such legislation. They included persons who had lost employment during the pandemic and found that those living in Medicaid expansion states were less likely to have moderate to severe mental distress following their job loss compared to those living in non-expansion states [40]. Another US study examined the effect of the Affordable care act (2014) on eligible women. Using repeated cross-sectional data the study found a lower proportion of women reporting post-partum depression after the introduction of this legislation [41].

**Work incentives and employment.** Gregg et al. [42] showed that a series of welfare reforms introduced in 1999 in the UK (including in-work tax credits and welfare-to-work programmes) was associated with a decrease in psychological distress among single mothers but not in the two control groups (married and single women). Explanatory analysis indicated that the difference in trends was accounted for by markers of longer-term financial deprivation rather than employment. Harkness examined the same UK welfare reforms but restricted the categorisation of single mothers to those that had been single for more than a year in order to account for 'transition effects' on mental health in newly single mothers. Results indicate that being in work was associated with a significant, 12.1% reduction in psychological distress among single mothers while the effects for married mothers were not statistically significant. Overall, working lone mothers had no greater risk of psychological distress than partnered mothers [43]. In another UK study, Barr et al [44] found that reforms that re-evaluated eligibility of out-of-work disability welfare benefits (implemented in 2010) were associated with higher anti-depressant prescriptions and higher self-reported depression at the regional level. Analysing the impact of the same reform but using individual-level data Curnock et al [45], showed that transitioning from receiving disability benefits into either paid work or to 'unemployment status' was associated with improvement in overall mental health: scores on the SF-12 increased by 5.9 points and 3.1 points respectively.

Katikireddi et al [46] examined the impact of the 'Lone Parent Obligation' (LPO) reform which entailed reducing income support and mandated seeking work for welfare once women's youngest child reached a threshold age. They found lower mental health scores in parents obliged to seek work compared with the group not exposed to reform (i.e., whose children were younger than the threshold age).

Wickham et al. [47] investigated the impact of Universal Credit reform, which was intended to provide greater incentives for claimants to enter employment. The results indicated higher psychological distress and poorer mental health in those eligible for Universal Credit after implementation of the reform. Williams [48,49] also examined the mental health impacts of restrictions on unemployment benefit (Job Seekers Allowance), in the form of benefit sanctions, implemented between 2010 and 2012. The results indicated that for every 10 additional sanctions applied per 100,000 population the rate of antidepressant prescriptions was 1.74 items higher (2019 study). Analyses in a larger number of districts showed higher rates of self-reported anxiety and depression (2021 study). For both outcome measures (antidepressants prescriptions, self-reported depression), a further increase was observed after the sanction policies became more severe (from end of 2012): every 10 additional sanctions applied per 100,000 population were associated with 4.57 additional antidepressants prescribing items ($p < 0.001$).

We included three US studies on welfare reforms that aimed to move single mothers from welfare to work. Jagannathan et al. [50] studied the New Jersey Family Development experiment, showing that women in the intervention, which stressed welfare-to-work experienced higher odds of clinical depression (based on Medicaid claims) compared to controls that continued to receive traditional welfare benefits. Rote et al [51] studied the Personal Responsibility and Work Opportunity Reconciliation Act of 1996 (PRWORA) which requires mothers to work in return for assistance and limits total federal lifetime TANF eligibility to 60 months. After reform, welfare recipients reported significantly more depressive symptoms than other women living in poverty when compared to the period before reform. Morris [52] also examined health impacts of the PRWORA by focussing on whether emphasis on quick job entry and/or personal attention in making the transition to employment made a difference in depressive symptoms. Clients of programs in the highest quartile of emphasis on quick job entry, particularly those with preschool children, experienced higher depressive symptoms based on CES-D score, when compared to the control group (clients not subject to changes in welfare package). Personalised support by caseworkers had no effect on depressive symptoms.

**Parental leave policies.**  We found nine studies that examined the impact of parental leave policies, all published in the past eight years. Most studies leveraged changes in maternal or family leave policies as a natural experiment.

We included four studies from the US. In a longitudinal study among 72 couples expecting their first child, Cardenas et al. [53] found that mothers whose partners took paid paternity leave had smaller prenatal-to-postpartum increases in depressive symptoms, whereas no difference was observed in the fathers themselves. Three studies examined the impact of implementation of paid family leave policy in California, which was the first US state to implement such a policy, in 2004. In a difference-in-difference analysis, all three studies [54–56] showed reduced psychological distress levels after the implementation of that policy, both among mothers and fathers.

Australia adopted a national paid parental leave policy in 2011. Two studies showed improved mental health among mothers after the policy had been implemented [57,58]. In addition, Bilgrami et al. [57] showed that the impact on mental health was larger among those mothers whose partners had access to a complementary policy specifically for partners.

The three European studies showed similar results. Avendano et al. [59] used data from the SHARE study, a panel survey of a representative sample of the European population aged 50+, finding that paid maternity leave in seven European countries was associated with lower depressive symptoms in older age. The other two studies were designed as natural experiments, and studied the impact of policy changes in Norway and Denmark [60,61]. In Norway, a national paid maternity leave arrangement was implemented in 1977 and Bütikofer et al [61] found improved maternal mental health after the introduction of this policy. On the basis of complementary research on explanatory variables, the authors suggest that improvements were mainly driven by the mother spending more time at home after childbirth [61]. A Danish study on the impact policy reform that increased in the length of maternity leave (from 24 to 46 weeks of fully compensated leave) did not show a reduction in hospitalisation due to depression or use of anti-depressant medication among mothers [60].

**Pension policies.**  We found two studies that examined the impact of increasing pension age. In the Netherlands, de Grip et al. [62] found that a 2006 reform that increased pension age from 62 years and three months to 63 years and

four months had a sizable impact on depressive symptoms: two years after the policy change high depressive symptoms were about 40% higher in participants exposed to the reform as compared to those not exposed. This was particularly so in participants that experienced a larger income loss because of the reform. In a UK study among women Carrino et al. [63] found that a reform that gradually increased the State Pension Age from 60 to 66 years within a short time-frame (10 years) led to an increased the likelihood of psychological distress and poor mental health. This effect was confined to women in jobs with high demands and low control.

In a Korean study, Kim et al. [64] examined the introduction of a Basic Old-Age Pension (BOP) in 2008, which provided a fixed pension for older persons in the lowest income category considering household income and assets. The association between receiving the BOP and depressive symptoms was not statistically significant, however, the authors reported that although the BOP improved income, it was not substantial enough to resolve poverty or relative deprivation, which may explain the lack of effect observed.

**Housing benefits.**  A UK study by Reeves et al. [65] evaluated the impact of cuts in the Government Housing Benefits - a programme which provides funds for tenants who rent housing in the private sector - on depressive symptoms in low-income households. The study found that the financial cuts to the housing allowance scheme (approximately 2,300 USD per year) significantly increased the prevalence of depressive symptoms by 10%.

**Childcare benefits.**  A Canadian study by Lebihan and Tangkomo [66] examined the effect of the Universal Child Care Benefit which provided for an unconditional cash transfer to families with children aged up to 6 years. The study found a small but non-significant improvement on depressive symptoms with indications of a stronger beneficial effect in families with low parental educational level and in families with girls.

**Public transportation policies.**  Reinhard et al. [67] examined the impact of the introduction of free bus passes to adults aged 60 years and older. Results showed that public transportation use increased after introduction of the policy and the policy was associated with a reduction of depressive symptoms. Based on the results on the association between transport use and possible explanatory factors the authors suggest that the benefits may stem from reduced loneliness, increased participation in volunteering activities and increased contact with children and friends.

**Educational policies.**  We found only one study that examined the impact of years of education on depressive symptoms. Hamad et al. [68] took advantage of variation in United States state-level compulsory schooling laws, a natural experiment that was associated with geographic and temporal differences in the minimum number of years that children were required to attend school. They found that increased years of education was associated with lower depression rates based on CES-D-8.

**Environmental noise exposure.**  Two studies examined the effect of environmental noise exposure. In a US study Wang et al observed an increase in depression diagnoses (based on Medicaid claims) in two New York neighbourhoods that were subject to increased air traffic noise due to redirection of flight paths [69]. A study in the Netherlands examined changes in depressive symptoms in an older population after introduction of policy to reduce aircraft noise levels. The authors observed that the policy failed to achieve its goal and there was no change in population-level symptoms [70].

## Study findings: Material resources

Following data extraction, the included studies on material resources were classified into 4 categories: income, employment, housing and collective material insecurity (Global Financial Crisis). The findings are shown in Table 2.

**Income supplement.**  Costello et al [71,72] evaluated the effect of increase in income due to a newly opened casino on a native-American reservation on mental health of children. Residents profited in terms of paid jobs and through receiving a fixed income in the form of percentage of the generated profits. There was no significant difference in depression or anxiety outcomes either 4 or 10 years after the opening of the casino, although children whose families moved out of poverty (14% of all children) had significantly fewer behavioural problems. Another US study by Courtin

et al [73] examined the effects of a conditional cash transfer to families, contingent to spending money on education, preventive healthcare and parental employment, but found no effect on psychological distress.

**Income volatility.** Income volatility refers to the year-to-year change in income for an individual or a household. The majority of studies on income volatility reported deterioration in depressive symptoms with reduced income.

Prause et al [74] found that downward income volatility was associated with higher depressive symptoms over a 6-year follow up period. However, this association was not significant when absolute volatility (regardless of the direction of change) was low. The authors propose that downward income change, rather than change per se, affects depressive symptoms. This view is consistent with findings of Sareen et al [75], Benzeval et al [76], Barbaglia et al [77] and Lorant et al [78], which found that a decrease in income was associated with depression incidence, psychological distress and depressive symptoms respectively. Two studies focused on poverty transitions. Wickham et al [79] found that transitioning into poverty, defined as <60% of national median household income, was associated with increased psychological distress. Thomson et al [80] found that the negative effects on psychological distress when transitioning into poverty were stronger than the positive effects of transitioning out of poverty. Several studies examined changes of financial status within the context of the 2008 Global Financial Crisis (GFC): Lindstrom et al [81] found that worsened financial status after the crisis was associated with higher risk of psychological distress, whereas improved financial status was not; Swift et al [82] found that income drop was associated with higher depressive symptoms. Curl and Kearns found that increased financial difficulties, in relation to austerity measures post GFC, was associated with reduced mental health [83].

One study examined the influence of loss in wealth. Within the context of the GFC, McInerney et al found that reduced wealth was associated with increased depressive symptoms and use of antidepressants among adults aged 55 years and older, with these effects being larger among respondents with high levels of stock holdings prior to the crash [84].

No association with changes in income were reported by two studies. McCarthy et al [85], included US and Canadian participants and found no association between changes in income with either depressive symptoms or mental health, although this effect was mediated by material hardship. McKenzie et al [86], in a study of changes in socio-economic status found no evidence that reduced household income was associated with psychological distress or mental health.

**Employment.** Studies on employment focused on loss/gain of employment, reduction of working hours or changes in employment quality. Overall, gaining employment was associated with lower depressive symptoms over time [87–96] while losing employment or having working hours reduced was associated with increased risk of depression [77,86,91,97–99]. Similarly, low quality employment trajectories, such as job instability and cumulative disadvantage were associated with higher risk of depression [100–103]

Studies looking at the effects of employment on depression reported some demographic differences. Lahelma [96] found that registered job-seekers who obtained a paid job had four times lower the odds of psychological distress than those who remained unemployed; this effect was stronger in men than in women. Differential effects based on gender were also observed by Jonsson et al [102], Hoven et al [101] and Kim at al [99]. Ali & Avison [104] found that transition either in- or out- of paid work was not associated with changes in depressive symptoms for the combined sample but losing work was associated with increased depressive symptoms in single compared to married mothers. Two studies by Eisenberg-Guyot et al in the US and Brudsten et al in Sweden and observed that ethnic minorities were more likely to experience adverse employment trajectories, which in turn was associated with reduced mental health [100,103]. Other studies reported that psychological resources and financial strain partly explained findings in single mothers transitioning out of work [105–107]. Zabkiewicz et al [106,107] examined the effects of gaining employment in single mothers receiving Temporary Aid to Needy Families (TANF), which carries a condition that recipients seek work. The authors found that gaining employment was associated with lower depressive symptoms, but effects were not observed in substance users or in women with more than three children. Breslin and Mustard [108] found that becoming unemployed was associated with higher risk of depression over a timespan of two years, but only in adults aged 31–55 years. Using data from an older population, Gallo et al. [109,110] investigated whether involuntary job loss due to a business closing or layoff was

associated with changes in depressive symptoms and found that in the long term (4–6 years) the initial association with depressive symptoms was only present in those with low financial means (2006 study). Only two studies, Lorant et al. and Salm [78,111] reported no association between becoming unemployed and depressive symptoms.

**Job security.** Two studies considered the impact of the threat of job loss. In a study of British Civil servants, Ferrie et al. [112] showed that those reporting changes in job security (but not job loss) over a 2.5 year period had higher psychological distress compared to consistently job secure participants.

A Swedish study by Magnusson Hansen et al. [113] indicated that risk of dismissal was associated with higher depressive symptoms, particularly for participants reporting consistent threat. The authors also found evidence that depressive symptoms influenced how a person perceived threats of dismissal.

**Housing.** Studies considering housing included multiple components such as quality and affordability, both at the individual level as at the neighbourhood (collective) level. At the collective level, exposure to neighbourhood characteristics might change either because of improvements in the environment, or as a result of residents being relocated to a more favourable neighbourhood.

**Housing Quality at the individual level: Adequacy.** A UK study by Blackman et al. [114] studied the health impact of reallocation of social housing based on medical needs. The study found larger improvements in mental health in participants that obtained housing conform their needs compared to participants still on a waiting-list, but no differences in the use of antidepressants. Petersen et al. [115] found that a selective licencing scheme, which aimed to improve rental housing quality was associated with lower area-level mental healthcare use. Pevalin et al found that persistently living in poor housing was associated with poor mental health and that having experienced this over a longer timeframe had effects over and above the current housing situation [116].

**Housing at the individual level: Affordability and stability.** In the US, Denary et al. [117] studied the effect of affordable housing in low-income adults by comparing those receiving rental assistance with those either wait-listed or not receiving assistance. The study found decreased psychological distress among participants that obtained rental assistance, but this was not statistically significant.

Three U.S. studies found increased risk of depression with housing eviction [118–120]. Conversely, another US study, conducted during the covid-19 pandemic, found that states with strong moratoriums on housing eviction had lower prevalence of depressive symptoms. This effect was stronger in Hispanic and non-Hispanic white populations that in African Americans [121]. A UK study by Prevalin [122] found a positive association between housing repossession (i.e., in the case of home ownership) and psychological distress. Participants evicted from rental properties showed a slight increase before the event but this effect did not persist. Using the same study population, Taylor, Pevalin and Todd found that going into payment arrears (either rent or mortgage payments) was associated with psychological distress [123]. This was also observed in a study by Alley et al, that examined mortgage delinquency [124] and by Kim et al, when housing became unaffordable [125]. The latter study observed that depressive symptoms reduced in those who moved into affordable housing [125].

Finally, instability in housing, characterised by having multiple addresses over time was associated with higher depressive symptoms in two US studies [126,127].

**Housing at the collective level: Economic (dis)advantage.** Leventhal et al. [128] studied the "Moving to Opportunity" experiment in New York (US) to assess the to assess the effect of moving from a high-poverty to a low-poverty area in families with children. Parents that moved to low-poverty neighbourhoods reported fewer depressive symptoms than those who remained in high-poverty neighbourhoods.

Cooper et al [129] assessed the effects of a housing relocation programme in residents of public housing and found that moving to an area with less economic deprivation was associated with reduction of depressive symptoms. McKenzie et al [86] reported no association between change in area-level deprivation score and psychological distress over a five-year period among a general sample of the population.

**Housing at the collective level: Urban regeneration programmes.** Eleven studies examined the effect of multi-component urban regeneration programmes on the mental health of residents. Harvey [130] used a pre-post design to examine changes in mental health following housing improvements in a Scottish neighbourhood and found 15% reduction in psychological distress following regeneration. Huxley et al. [131] examined the impact of the Single Regeneration Budget area in Manchester and found that the prevalence of psychological distress did not differ between the intervention versus control neighbourhoods at follow-up. Although some improvements had been observed in the intervention areas, the authors noted that little of the available budget was spent, which might explain the lack of an effect.

Rose et al examined the Community Wealth Building programme in one of the 20% most deprived local authorities in the UK. The programme was designed to boost economic development through, for example, improved employment conditions and support of local supply chains by public and non-profit organizations. This resulted in reduced antidepressant prescriptions and prevalence of depression relative to the control areas [132].

The UK New Deal for Communities (NDC) programme (1999–2011) was studied by Stafford et al. [133] and Walthery et al [134]. The programme targeted 39 of the most deprived areas in England, with activities in multiple domains (crime, physical environment, community etc.). Stafford et al observed a decline in psychological distress in some intervention areas between 2002 and 2008, however, the difference with comparator areas was not significant. Using longitudinal data from each of the NDC areas, Walthery et al. also did not find evidence that the programme impacted mental health, although respondents in poorer socio-economic circumstances in NDC areas fared better than those in comparator areas.

White et al. [135] and Greene et al. [136] report on the impact of 'Communities First', an area-wide regeneration programme delivered in the 10% most deprived areas in Wales. The programme addressed multiple themes, such as crime, education, and building community facilities. After the intervention period, mental health in the intervention group was better than in comparable areas. Mediation analysis [136] indicated that over half of the observed effect was explained through improvements in neighbourhood quality (littering, nuisance from dogs and noise: 22%), reductions in disorder (vandalism, crime rates and discarded needles and syringes: 19%) and sense of neighbourhood belonging (11%) and, for a small extent, by social cohesion (1.7%).

Egan et al [137,138] assessed effects of an urban renewal programme in 14 deprived neighbourhoods in Glasgow, Scotland. The programme, addressed housing quality, as well as social factors (e.g., debt payment services, playgrounds, employment support). Using data from 2006–2008, they found small improvements in mental health among residents experiencing housing improvement compared to residents of similarly deprived non-target areas. A follow-up study from 2006 to 2011 found that improvements in mental health were higher in areas with greater financial investment compared to lower investment areas.

Two studies in the Netherlands examined the impact of an urban regeneration programme, implemented between 2008 and 2011 in deprived neighbourhoods. The programme included multiple interventions, such as creation of green space and facilities for sport, improving safety and social cohesion. Jongeneel et al. [139] found that the trend change in mental health between the intervention and pre-intervention period was approximately the same in the target districts as in deprived control districts. Improvement in mental health was found in those target districts that implemented a more intensive programme. A study by Timmermans et al. [140] in a cohort of older participants found no evidence of an effect of this programme on depressive symptoms.

**Collective material insecurity: Financial crisis.** The impact of the Global Financial Crisis (GFC) of 2007/8 and subsequent austerity policies on mental health has been examined in several countries; Hong Kong, Greece, Ireland, Spain, France, the US, UK and Canada and across Europe. Only three studies found no changes in population-level depressive symptoms post GFC [141–143].

Several studies examined the specific conditions under which risk of depression was increased. In addition to the studies described in the section on 'Income volatility', earlier in table 2 [82–84], Lee et al. [144] found that the prevalence of depressive episode was higher in those experiencing loss in financial investments. Madianos et al. [145] and Economou

et al [146] found higher depression in cases of financial hardship or in people who experienced higher economic distress. Modrek et al. [147] examined changes in utilization of mental health services and medications in a cohort of continuously employed (and insured) workers of a manufacturing firm in the US that experienced significant downsizing events during the GFC. The results showed that workers used more antidepressants after the recession (13%), which contrasted with a decreasing trend of use before the recession and that this effect was stronger in high lay-off plants.

In a study across 20 European countries, Buffel, Van de Velde and Bracke found increased prevalence of depressive symptoms in countries with increased unemployment rates post GFC, taking the economic state of each country before the crisis into account but found that these contextual effects were not fully explained by differences in individual-level employment [148]. Drydakis et al [149], Bartoll et al [150], Gili et al [151] observed an important role of individual-level unemployment in the higher risk of depression. Barr et al reported that increased area-level unemployment and decreased area-level income explained a large part (36%) of the decline in mental health observed post GFC [152]. The role of austerity measures was examined by two studies, Nour et al. [153] initially found no association between exposure to GFC and self-reported depression diagnosis by a physician but an increase herein after subsequent implementation of austerity measures in 2011–13; Cherrie et al. [154] showed that reduced area-level employment levels were associated with increased incidence of antidepressant use and that austerity measures such as reduced eligibility for welfare payments, and subsequent income reduction explained a large proportion of the observed effect. Two studies looked at ethnic differences in the effect of the GFC: An Irish study by Villarroel et al observed that depression post-recession was only increased in Irish men but not in non-Irish or African-origin men [155]; Gotsens et al observed that the development in mental health of immigrant women (but not men) was less favourable than among Spanish-born residents [156].

Finally, in an older study, of the 1991 economic recession in Sweden, Rahmqvist and Carstensen observed higher prevalence of psychological distress compared to pre-recession years. This effect was observed in both employed and unemployed persons [157].

## Study findings: Social resources

Following data extraction, the included studies on social resources were classified into two categories: social disorder (n = 4), and social participation (n = 7). See Table 3.

**Social disorder.**  Social disorder represents elements that result from the behaviour of people living in a particular environment such as safety, crime, violence, graffiti etc [158]. A US study by Cooper et al. [129] showed that relocation to neighbourhoods with lower social disorder (operationalised by violent crime rates and density of alcohol outlets) as part of a public housing relocation programme was associated with a persistent reduction in depressive symptoms during follow-up in both men and women. The change seemed to be driven by changes in perceived crime. Another US study by Mair et al. [159] found that increased neighbourhood safety, social cohesion and aesthetic quality were associated with lower depressive symptoms while increases in neighbourhood violence and stress was associated with increased depressive symptoms although the small sample size meant wide confidence intervals for all indicators.

Using longitudinal data on exposure to neighbourhood crime rates (non-domestic violence, malicious damage to property, break/enter, stealing/theft/robbery), Astell-Burt et al. [160] reported that an increase in neighbourhood crime was associated with greater psychological distress. Effect sizes were particularly high for women, especially when an increase in malicious damage was observed in the neighbourhood. In the UK, Dustmann & Fasani [161] examined the effect of local crime rates on psychological distress, using two large British panel surveys. Their findings showed a statistically significantly negative impact of overall local crime rates on psychological distress, especially in women.

**Social participation.**  This category includes studies that looked at different aspects of social interaction between individuals.

An Australian study by Cruwys et al. [162] examined whether depressive symptoms can be prevented by increasing social contact and facilitating social identification. Fifty participants at risk for depression (previous diagnosis of mental

illness) joined a community recreation group; joining a social group was associated with a reduction in depressive symptoms among participants who reported identifying with the group. A study in Ireland by McGale et al [163] examined the effect of promoting social support within a supervised exercise intervention, compared to supervised exercise alone or a group that had access to unsupervised exercise facilities. The study found that engaging in supervised exercise was associated with reduced depressive symptoms but there was no additional effect in the group that received the social support component.

Three of the included studies were conducted in Japan. Murayama et al. [164] examined the effect of reading books to young children at school on elderly persons' depressive symptoms. The study found that participation was positively associated with sense of manageability and meaningfulness; which in turn was negative associated with depressive symptoms. Watanabe et al. [165] examined whether municipal-level social capital was associated with depressive symptoms. Improvement in ten out of the 14 indicators of social capital were associated with a decline of prevalence of depressive symptoms.

Finally, a UK study by Lindström & Giordano [81] examined the impact of changes in social capital in relation to psychological well-being against the background of the GFC, with social capital operationalised as level of trust and social participation. The results indicated that individuals with low levels of trust in 2008 had increased psychological distress in 2008 compared to 2007, even after considering individual perceptions of financial strain.

Table 4 provides a summary of the main findings based on the categorisations applied.

## Discussion

In this review, we aimed to examine whether changes in social determinants can contribute to the prevention of depression, to provide input for policy development and to highlight research gaps. We found most, consistent and high quality of evidence for changes in a number of social determinants, where a *positive* development led to a *reduction* on risk of depression 1) paid parental leave (8/10 studies) 2) gaining employment (9/10 studies); or, conversely, where a *negative* development led to *increased* risk of depression 1) reduction in income or transitioning into poverty (12 studies), 2) losing employment (14/17 studies), 3) work incentives coupled with reduction/loss of welfare ((8/12 studies), 4) collective insecurity (economic crisis) (14/18 studies), instability of housing (9 studies). We present the main findings according to our categorisation of whether interventions addressed societal arrangements, material resources distributed through these arrangements, and social resources that follow from interactions between people. In order to streamline the discussion we make use of 'risk of depression' as umbrella term to cover all outcomes specified in the findings.

### Societal arrangements

The majority of studies on societal arrangements assessed the impact of welfare reforms. All of the studies can be considered high quality, i.e., > 18 points on the Validity Assessment scale. Of those, five studies scored 19–20 points. Most studies examined *restriction* of entitlements to social welfare and found that restriction of entitlements was associated with increased risk of depression. This finding is largely consistent with another recent review [10]. However, some studies reported that reforms in work incentives and employment also resulted in a *reduction* in depression risk. In the latter group of studies, changes in entitlements were accompanied by a wider package of incentives such as tax credits [42] or transitioning to another category of entitlement [45]. The studies on expansions of welfare entitlements, such as tax credit policies, showed either reduction in risk of depression or no effect (in the short-term). In this category and consistent with a recent review by Heshmati et al, paid parental leave policies most convincingly showed a positive effect, in particular among mothers [166]. The impact on fathers was less studied.

We found four studies that evaluated minimum wage policies and only one study each looking at educational policies, public transport policies, and environmental policies, limiting our ability to generalise these findings.

**Material resources distributed through these arrangements**

The determinants for which we found consistent evidence were generally studied in a range of countries. Becoming unemployed, for example, was related to increased risk of depression and was observed by studies in the US, Norway, Netherlands, Canada, UK, Spain and New Zealand, Sweden, South Korea, Japan, France, Belgium, Germany, Finland and Australia. Study quality was mixed, of the 28 studies related to employment, 11 scored less than 20 and of these five scored less than 18 points. Thus our findings regarding employment may be biased, although they were similar to those of previous reviews and meta-analyses [167,168]. The association between employment and risk of depression was consistent over time but may have changed in recent years due to changes in the labour market. For example, an ageing workforce, technological advancements, and economic shocks such as the global pandemic as well as the rise of the gig economy [169]. Ten of the 26 studies identified in our review were conducted post-2010, potentially capturing the effect of these changes, particularly the studies examining labour market trajectories/precarity [98,100–103]. However, the design of these studies means it is not possible to identify the conditions in the labour market that impact on the association between employment and mental health. This indicates a need for additional studies on the mental health impacts of (un) employment, that clarify the conditions under which employment can contribute to reducing risk of depression.

Evidence for the effect of income loss was much stronger than for a gain in income, although study quality was generally lower than for other determinants studied with 6/12 studies scoring higher than 20 points. Our finding is consistent with the results of a meta-analysis that also included low- and middle-income countries [21]. The negative effect of income loss was found to be stronger in those with lower income levels or close to poverty, as is also discussed by Shields-Zeeman and Smit [22]. In our review only two of the seven studies found this negative effect of income to be specific in lower income groups [78,79] implying that in the case of high-income countries the underlying mechanism for this effect is not restricted to absolute poverty. It is likely that experiencing relative poverty might act, for example, through perceptions regarding societal standing (status anxiety) or stigma associated with being poor.

Studies on housing at the collective-level, through urban regeneration showed mixed results. About half of these studies indicated no positive impact on risk of depression. The other half indicated an impact, primarily within subgroups (e.g., women) or in areas with higher realised investment.

**Social resources that follow from the interaction between people**

This was the least studied set of social determinants. Improvement and deterioration of social disorder at the neighbourhood level was consistently associated with a lower and higher risk of depression respectively. Three of the five studies on social participation showed that increase therein was followed by a reduction in risk of depression, although study quality was low, with low numbers of participants

**Determinants with limited evidence**

For a number of social determinants examined by studies in this review, we had insufficient evidence to draw a conclusion on its (in)effectiveness. This specifically applied to expansions in access to social welfare other than paid parental leave policies, societal arrangements other than welfare reforms, increased income, urban regeneration programmes, and social participation. The reasons for the insufficiency of the evidence varied from a limited number of studies with limited variation in settings (e.g., expansions to social welfare through Earned Income Tax Credit programmes or access to health insurance), mixed results without having insight into the reasons for this (e.g., urban regeneration programmes) or too limited to draw a conclusion (e.g., social capital). For housing security, the results were inconsistent across spatial levels; at the individual level, studies consistently indicated a change in risk of depression, negative and positive following deterioration or improvement respectively of housing insecurity, affordability or threat of eviction. Studies at the neighbourhood level, including urban regeneration yielded mixed results which cannot be ascribed to low study quality; only one of the 11 studies scored <18 on the Quality Assessment scale. The difference in results between these two types of studies may be

due to the fact that in studies at the individual level, the exposure consisted of a clearly defined improvement or deterioration, e.g., housing eviction, whereas the change in exposure at the neighbourhood level, in urban renewal programmes, was less clearly defined, often including multiple unspecified changes. Interestingly, studies aimed at the social and community environment with a more specific focus, such as crime levels [128,129,160,161], social cohesion [159,165], gave a more consistent picture, i.e., improvements in depression risk with improvement in exposure.

## Strengths and limitations

The focus of this review was on depression. However, the majority of the studies included measure more general constructs like psychological distress or mental health. There is a dearth of research on the impact of changes in social determinants on rates of depression at the population level. While this information is highly relevant, obtaining such data is highly challenging. Moreover, the field is currently moving more towards broader outcomes such as wellbeing and functioning. In order to capture as many studies focused on intervening on a social determinant as possible we chose to expand our inclusion to these broader outcomes.

We aimed to develop an overview of potential policies and intervention that act on social determinants of mental health relevant for high income country contexts, and we may have missed specific mechanisms at play in low and middle-income country contexts. Excluding low and middle-income countries in our search means that we might have missed capturing the mechanisms underlying the association between these determinants and depression. However, the influence of the social determinants can be very context-specific. For example, the level of economic development is likely to influence the impact of such interventions, especially progressive policies like parental leave, although there is likely to be variation in contextual factors between high-income countries as well.

Our search strategy was very broad and we did not include a limitation on publication date to avoid missing older studies. The broad inclusion of determinants means that we were only able to conduct a qualitative synthesis of the included papers and relied on authors' reporting for any quantitative evaluation of findings in terms of meaningful effects. We also may have missed some relevant articles, as our search terms may not have been exhaustive for some topics, such as social participation, which can be operationalised in a variety of different ways, including social cohesion or social capital [12]. We conducted our search in three major databases (Medline, Embase and Psychinfo) so some relevant papers may have been missed. To compensate, we conducted a thorough search of the reference lists of included studies and published reviews. Finally, we restricted our focus on studies that examined change in both determinant and outcome, which likely explains the limited numbers of studies included on some topics, specifically, on social resources which has been the topic of other reviews [170,171]. Nonetheless, we believe our approach helped capture the state of the art with regards to the potential for interventions to prevent depression across a broad range of social determinants.

## Implications for further research

Limiting our search to studies that included change in both determinants and outcomes was useful in expanding our understanding of what can be expected from interventions addressing social determinants; as not all associations between such determinants and depression are causal. Consequently, a broad range of determinants, identified in observational studies, has not been included in this review. For example, social norms towards mental illness, social pressure, status anxiety, excessive use of social media, exposure to noise, green/blue space etc [6,12,172,173]. This emphasizes the need for studies with similar designs as included in our review on a much broader range of social determinants than currently studied.

The included studies used a variety of instruments and outcome measures, some of which specifically assessed depressive symptoms. An important area for consideration in future research is to ensure that large cohort studies, also those that include clinical assessment of depression, expand measures to include a variety of social and economic risk factors, either by adding these variables to their data collection process or linking their data to other databases on (changes in) social determinants.

As previously mentioned, studying change in the social determinants of health is fraught with difficulty, e.g., populations cannot be randomized to a particular condition to adhere to the gold standard of a randomized controlled trial (RCT). A number of the included studies used natural experiments or instrumental variables where an RCT may not have been possible or appropriate. However, the application of such methodologies remains limited and could be explored more by researchers as could the potential of applying new causal inference methods in the evaluation of interventions. Further, although many authors discussed potential mechanisms underlying the relationships observed few studies included in our review employed formal mediation analysis to try and unravel such mechanisms which indicates the need for more mechanistic studies in this field.

Although alternative experimental designs, causal inference and mediation analysis help expand our understanding of the role of individual social determinants in depression, they do not reflect complex reality, where multiple factors act at different levels of influence, ranging from the individual- to the social- and environmental-levels. This need is perhaps mostly clearly illustrated by the studies evaluating widely divergent, multi-component interventions (i.e., urban renewal interventions), studies with differential effects depending on population characteristics and in studies with no clear direction of effect, such as studies of policies to incentivise employment. To capture this complexity, there is a need to apply systems approaches [174] and realist approaches [175], focusing on questions such as: under which conditions do changes in social determinants influence depression and for whom; how do different determinants of mental health/depression (e.g., social cohesion and economic improvement in a neighbourhood or in the urban environment interact in their impact on mental health? In taking this approach it is important to include consideration of the role of political, economic, or logistical constraints in adoption of policy to address the social determinants of health.

## Implications for policy

Traditional approaches to improve mental health or prevent depression often target the individual and their role in shaping their own outcomes. This focus on the individual suggests that the responsibility to overcome barriers such as poverty lies with the person rather than with allocation of resources by governments and institutions [176]. Focusing on social determinants as a way to impact depression levels proceeds from the premise that even relatively small shifts in the distribution of depressive symptoms at the population-level could be expected to have large benefits for the population as a whole [177]. This is relevant considering the relatively high societal and healthcare cost of even mild depressive symptoms [5,178].

We found consistent evidence for policies aimed at providing paid parental leave; promoting paid employment; and preventing income loss/financial distress. Importantly, our review has shown the importance of mitigating financial insecurity (through austerity measures) in a context of financial crisis, as well as preventing job insecurity.

Our review also highlights the importance of considering the unintended consequences of policies. For instance, reforms that incentivise employment may impact other factors the influence mental health. The Job Seekers Allowance in the UK [47–49] and the introduction of the Personal Responsibility and Work Opportunity Reconciliation Act in the US [51,52]), were both consistently followed by an increase in depression risk, probably through their restriction of welfare benefits. Other negative effects of incentivising work maybe due to increased need for childcare or insecurity due to having to take on menial, unstable work.

The lack of clear-cut evidence on many of the other determinants included in this review was due to limited numbers of studies or methodological issues, thus we cannot conclude that they are not relevant for policy.

Overall, the broad spectrum of actions needed to address population-level depression risk implies there is a need for a comprehensive approach that involves collaboration between different policy domains. In addition, the complexity of addressing population mental health, given interactions between different determinants, differential effects across populations and unintended consequences implies careful consideration of the local context in developing policy, preferably using system dynamic approaches.

## Conclusion

This review builds upon reviews of observational and longitudinal studies that show an association between social determinants and risk of depression. We wanted to extend that evidence by studying whether this association can be influenced/changed through policy or intervention and to identify gaps in evidence. Overall, we found evidence that population-level depression risk can be reduced by policies ensuring the provision of paid parental leave; prevention of income loss/financial distress; and the promotion paid employment. Whereas reduced or conditional entitlements to social welfare, loss of income and financial distress, loss of employment and collective insecurity can increase depression risk. We argue that there is a need for studies on a broader set of social determinants using systems and realist approaches to understand the complex interactions between determinants and contextual factors.

## Supporting information

**S1 Data. Search strategies.**
(XLS)

**S2 Data. Overview of included and excluded papers.**
(XLSX)

## Author contributions

**Conceptualization:** Mary Nicolaou, Karien Stronks.

**Investigation:** Mary Nicolaou.

**Methodology:** Mary Nicolaou, Laura S. Shields-Zeeman, Junus M. van der Wal, Karien Stronks.

**Writing – original draft:** Mary Nicolaou, Karien Stronks.

**Writing – review & editing:** Laura S. Shields-Zeeman, Junus M. van der Wal.

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
