## [Decision Letter · Decision Letter 0]

12 Feb 2024

PONE-D-23-23924Preventing depression in high-income countries – a systematic review of studies evaluating change in social determinantsPLOS ONE

Dear Dr. Nicolaou,

Thank you for submitting your manuscript to PLOS ONE. After careful consideration, we feel that it has merit but does not fully meet PLOS ONE’s publication criteria as it currently stands. Therefore, we invite you to submit a revised version of the manuscript that addresses the points raised during the review process.

We look forward to receiving your revised manuscript.

Kind regards,

Mohammad Hossein Ebrahimi

Academic Editor

PLOS ONE

Journal Requirements:

Reviewers' comments:

Reviewer's Responses to Questions

**Comments to the Author**

1. Is the manuscript technically sound, and do the data support the conclusions?

Reviewer #1: Yes

Reviewer #2: Yes

Reviewer #3: Yes

2. Has the statistical analysis been performed appropriately and rigorously? 

Reviewer #1: Yes

Reviewer #2: N/A

Reviewer #3: N/A

3. Have the authors made all data underlying the findings in their manuscript fully available?

Reviewer #1: Yes

Reviewer #2: Yes

Reviewer #3: Yes

4. Is the manuscript presented in an intelligible fashion and written in standard English?

Reviewer #1: Yes

Reviewer #2: Yes

Reviewer #3: Yes

5. Review Comments to the Author

Reviewer #1: in my opinion, the article was complete and comprehensive in every respect, ethical and scientific issues were observed.

Experiments have been conducted rigorously, with appropriate controls, replication, and sample sizes

the statistical analysis been performed appropriately and rigorously.

Reviewer #2: The manuscript undertakes systematic review on a very pertinent topic highlighting whether changes in social determinants can prevent depression and its symptoms in high income countries.

Some points can be worked upon for improvimg the overall rigour and quality of the study.

1. The authors have used the word 'sex" throughout the manuscript. The word 'sex' is a biological term, and it is therefore suggested that given the context of the manuscript, the word 'gender' is more suitable as it has social connotations.

2. The MeSH (Medical subject headings) terms are missing in the manuscript. Please include it in the search strategy.

3. The discussion is very exhaustive and it can be shortened slightly to make the manuscript more crisp.

Reviewer #3: Thank you for giving me the opportunty to review this manuscript. The aim of this review is to examine whether change in social determinants can result in depression prevention using a systematic review. Social determinants of health are essentially non-medical factors affecting health outcomes.

Study characteristics were clearly and concisely summarized. The presentation of the results were structured into societal arrangements, material resources, and social resources, with sufficient details. Table 4 (Summary of Findings) is an essential table for readers to grasp a better understanding of the relationship between change in social determinants and depression.

The following are the specific comments:

1. This study demonstrates a connection between social determinants and depression risk, highlighting several key factors that policymakers should take into account when developing policies or interventions to reduce population-level depression risk. While this review may not specifically address potential barriers or challenges in implementing such policies and interventions, such as political, economic, or logistical constraints, these factors are indeed crucial for policymakers to consider when making decisions. It may be worth discussing the potential barriers or challenges that could arise in implementing policies or interventions aimed at reducing depression risk.

2. Moreover it would be beneficial for policymakers to have information on the relative impact of these factors in reducing depression risk. This would enable them to prioritize policies and interventions, with the constraints of limited resources.

3. The review includes studies of various methodological rigor and quality (Validity Assessment scores ranging from 18 to 26). Although studies with a score of 18 or higher were considered to be of high quality, some with relatively lower score (eg 19) may indicate that the existence of potential biases or confounding factors in these studies, which could impact the strength and generalizability of the findings. Please explain how these scores be taken account in the discussion/conclusions.

4. This review does not provide specific details on the types of policies and interventions that may be effective in reducing population-level depression risk. This may limit the applicability of the findings for policymakers and practitioners seeking to implement evidence-based interventions.

5. The conclusion acknowledges the need for studies that consider different populations and conditions. This will be more useful to elaborate more on this point so that a specific guidance can be formulated on how to tailor interventions to different contexts or how to account for potential variations in effectiveness across settings.

6. PLOS authors have the option to publish the peer review history of their article (what does this mean? ). If published, this will include your full peer review and any attached files.

**Do you want your identity to be public for this peer review?** For information about this choice, including consent withdrawal, please see our Privacy Policy .

Reviewer #1: No

Reviewer #2: **Yes: ** TANVI KIRAN

Reviewer #3: **Yes: ** LM Ho

---

## [Author Response · Author response to Decision Letter 1]

23 Apr 2024

Financial disclosure

This study was funded by the Centre for Urban Mental Health, which is one of the research priority areas (RPA) of the University of Amsterdam. Specifically, the centre funded the salaries of MN (part-time) and JvdW (full-time).

Data availability

All data are in the manuscript and/or supporting information files. Specifically, this paper is a review of existing studies and, as such, does not include any unique data, the references to the included articles are part of the manuscript.

Supporting information

We have adjusted the reference to supporting information in the text to reflect PlosOne guidelines and also include mention of the files at the end of the manuscript.

Responses to reviewers comments

We appreciate all three reviewers’ positive comments on our manuscript.

Reviewer #1: in my opinion, the article was complete and comprehensive in every respect, ethical and scientific issues were observed.

Experiments have been conducted rigorously, with appropriate controls, replication, and sample sizes. the statistical analysis been performed appropriately and rigorously.

Thank you!

Reviewer #2: The manuscript undertakes systematic review on a very pertinent topic highlighting whether changes in social determinants can prevent depression and its symptoms in high income countries.

Some points can be worked upon for improving the overall rigour and quality of the study.

We thank the reviewer for their positive evaluation of our review.

1. The authors have used the word 'sex" throughout the manuscript. The word 'sex' is a biological term, and it is therefore suggested that given the context of the manuscript, the word 'gender' is more suitable as it has social connotations.

Adjusted as suggested.

2. The MeSH (Medical subject headings) terms are missing in the manuscript. Please include it in the search strategy.

We have included the MeSH terms used in supplement 1 where the full search strategy is described.

3. The discussion is very exhaustive and it can be shortened slightly to make the manuscript more crisp.

We have worked to shorten the discussion section by reducing some of the detail in the discussion. Additionally, we have considerably restructured the text, also in response to comments by reviewer 3. We hope that the reviewer finds the text to be less dense as a result. We refer to the text in lines 763-770; 804-859; 868-900; 1199-1224.

Reviewer #3: Thank you for giving me the opportunty to review this manuscript. The aim of this review is to examine whether change in social determinants can result in depression prevention using a systematic review. Social determinants of health are essentially non-medical factors affecting health outcomes.

Study characteristics were clearly and concisely summarized. The presentation of the results were structured into societal arrangements, material resources, and social resources, with sufficient details. Table 4 (Summary of Findings) is an essential table for readers to grasp a better understanding of the relationship between change in social determinants and depression.

We thank the reviewer for her/his positive evaluation of our review. We have also checked table 4 to ensure that it accurately reflects the included papers.

The following are the specific comments:

1. This study demonstrates a connection between social determinants and depression risk, highlighting several key factors that policymakers should take into account when developing policies or interventions to reduce population-level depression risk. While this review may not specifically address potential barriers or challenges in implementing such policies and interventions, such as political, economic, or logistical constraints, these factors are indeed crucial for policymakers to consider when making decisions. It may be worth discussing the potential barriers or challenges that could arise in implementing policies or interventions aimed at reducing depression risk.

We thank the reviewer for this comment. Based on this we have more strongly highlighted the issue of unintended consequences of policies (lines 1203-1212) and have mentioned the importance of inter-sectoral collaboration in forming policy, using systems approaches (lines 1219-1225). We have also restructured the discussion to ensure that the implications for policy are condensed in one section so that they are more accessible to readers (lines 1199-1224).

Beyond these changes, we have not elaborated on the barriers of challenges in implementing policies at this was beyond the aim of our review. To do so would require a different kind of approach than what we have taken. However, we do agree that this is an important issue to consider in future research and included the following comment in paragraph outlining the implications for research: In taking this approach it is important to include consideration of the role of political, economic, or logistical constraints in adoption of policy to address the social determinants of health. (lines 1184-1186)

2. Moreover it would be beneficial for policymakers to have information on the relative impact of these factors in reducing depression risk. This would enable them to prioritize policies and interventions, with the constraints of limited resources.

If we understand this comment correctly, the reviewer is suggesting that we rank interventions according to their relative impact. Our findings highlight a number of interventions that have been consistently shown to reduce population risk of depression as well as policy actions with adverse effects. To emphasise these we have added the following text in the first paragraph of the discussion: We found most, consistent and high quality of evidence for changes in a number of social determinants, where a positive development led to a reduction on risk of depression 1) paid parental leave (8/10 studies) 2) gaining employment (9/10 studies); or, conversely, where a negative development led to increased risk of depression 1) reduction in income or transitioning into poverty (12 studies), 2) losing employment (14/17 studies), 3) work incentives coupled with reduction/loss of welfare ((8/12 studies), 4) collective insecurity (economic crisis) (14/18 studies), instability of housing (9 studies). (lines 764-770).

However, we add that: The lack of clear-cut evidence on many of the other determinants included in this review was due to limited numbers of studies or methodological issues, thus we cannot conclude that they are not relevant for policy. (lines 1217-1219).

Thus we also add:

Overall, the broad spectrum of actions needed to address population-level depression risk implies there is a need for a comprehensive approach that involves collaboration between different policy domains. In addition, the complexity of addressing population mental health, given interactions between different determinants, differential effects across populations and unintended consequences implies careful consideration of the local context in developing policy, preferably using system dynamic approaches. (lines 1220-1225).

3. The review includes studies of various methodological rigor and quality (Validity Assessment scores ranging from 18 to 26). Although studies with a score of 18 or higher were considered to be of high quality, some with relatively lower score (eg 19) may indicate that the existence of potential biases or confounding factors in these studies, which could impact the strength and generalizability of the findings. Please explain how these scores be taken account in the discussion/conclusions.

We have included an evaluation of the evidence base on the VA scores in describing our main findings in the summary of our findings and in other sections of the discussion.

With regard to welfare reforms: The majority of studies on societal arrangements assessed the impact of welfare reforms. All of the studies can be considered high quality, i.e. >18 points on the Validity Assessment scale. Of those, five studies scored 19-20 points. (lines 779-781).

In relation to (un)employment: Study quality was mixed, of the 28 studies related to employment, 11 scored less than 20 and of these five scored less than 18 points. Thus our findings regarding employment may be biased, although they were similar to those of previous reviews and meta-analyses [167, 168]. (Lines 309-312).

Also: Studies at the neighbourhood level, including urban regeneration yielded mixed results which cannot be ascribed to low study quality; only one of the 11 studies scored <18 on the Quality Assessment scale. (lines 391-393)

4. This review does not provide specific details on the types of policies and interventions that may be effective in reducing population-level depression risk. This may limit the applicability of the findings for policymakers and practitioners seeking to implement evidence-based interventions.

As mentioned in our response above, we have restructured the discussion section to bring the evidence for policy together to make our findings more accessible. In this paragraph we emphasize the findings for which there is greater evidence but the goal of this review was not to rank policies, but to present evidence on policies which might have an effect on mental health. However, we also highlight the importance of considering local context and unintended consequences. For example, having paid parental leave has an impact on mental health of the family members and may also contribute to better mental health outcomes of the children later in life, but how that paid parental leave is embedded into policy measures (or combined) with other policies may be country specific. See also our response above.

5. The conclusion acknowledges the need for studies that consider different populations and conditions. This will be more useful to elaborate more on this point so that a specific guidance can be formulated on how to tailor interventions to different contexts or how to account for potential variations in effectiveness across settings.

We appreciate this comment, the relevance of context and potential unintended consequences is clear. We hope that the adjustments made in our manuscript are sufficient to meet the suggestions of the reviewer. Importantly, we want to emphasise that a comprehensive analysis on how to tailor interventions is beyond the scope of this review. We do suggest that further research, using systems and/or realist approaches might provide more specific insights for tailoring to context and population.

---

## [Decision Letter · Decision Letter 1]

24 Jun 2024

PONE-D-23-23924R1

Preventing depression in high-income countries – a systematic review of studies evaluating change in social determinants

PLOS ONE

Dear Dr. Nicolaou,

Thank you for submitting your manuscript to PLOS ONE. After careful consideration, we have decided that your manuscript does not meet our criteria for publication and must therefore be rejected.

Specifically:

I am sorry that we cannot be more positive on this occasion, but hope that you appreciate the reasons for this decision.

Kind regards,

De-Chih Lee, Ph.D.

Academic Editor

PLOS ONE

Additional Editor Comments:

There are some problems with this manuscript.

1. First, this manuscript only searches three journal databases, which may miss important information or references.

2. The literature included by the author consists of quantitative and qualitative research. The author conducts the discussion using a qualitative method. Using qualitative research methods to perform a quantitative literature review will miss much information and is unscientific and unobjective.

3. The author excludes patients with diabetes and heart disease. What about other illnesses such as cancer, mental disorders, etc.? What is the basis for the exclusion?

4. The authors exclude low-income and middle-income countries. Are these countries not providing social support? For example, China. When these countries are excluded, the conclusions of this manuscript will be biased.

5. In the abstract, the authors find evidence that strategies such as promoting paid employment and parental leave can reduce the risk of depression. What is the author's evidence? Is it through statistical analysis? I haven't seen any system evidence of this.

Finally, using qualitative methods for the literature review of quantitative studies was permissible in the past when tools were not available. However, many software tools are now available to assist researchers in conducting more objective and scientific analyses. In particular, this manuscript includes many references, most of which are quantitative. Using traditional manual methods to organize and summarise would have missed much important information and would not have been objective. In addition, the literature search and the inclusion criteria make this manuscript subject to serious bias and will miss important information or literature. It is recommended that the authors expand the literature search and adopt a quantitative method. Authors may include too wide a range of variables for quantitative methods can not be used. Authors are advised to focus on the research topic and expand the search of journal databases.

Reviewers' comments:

Reviewer's Responses to Questions

**Comments to the Author**

1. If the authors have adequately addressed your comments raised in a previous round of review and you feel that this manuscript is now acceptable for publication, you may indicate that here to bypass the “Comments to the Author” section, enter your conflict of interest statement in the “Confidential to Editor” section, and submit your "Accept" recommendation.

Reviewer #1: All comments have been addressed

Reviewer #3: All comments have been addressed

2. Is the manuscript technically sound, and do the data support the conclusions?

Reviewer #1: Yes

Reviewer #3: Yes

3. Has the statistical analysis been performed appropriately and rigorously? 

Reviewer #1: Yes

Reviewer #3: N/A

4. Have the authors made all data underlying the findings in their manuscript fully available?

Reviewer #1: Yes

Reviewer #3: Yes

5. Is the manuscript presented in an intelligible fashion and written in standard English?

Reviewer #1: Yes

Reviewer #3: Yes

6. Review Comments to the Author

Reviewer #1: in my opinion, the article was complete and comprehensive in every

respect, ethical and scientific issues were observed.

It would be better if the Cochrane site was also used in the search

Reviewer #3: Thank you for giving me the opportunty to review this manuscript again. This is a resubmission, and has significant improvements. It is particularly helpful to include a section on Determinants with limited evidence, as this enables readers to effectively summarize and appraise the presented evidence in this review. The authors have adequately addressed the concerns and comments that were previously raised by reviewers.

7. PLOS authors have the option to publish the peer review history of their article (what does this mean? ). If published, this will include your full peer review and any attached files.

**Do you want your identity to be public for this peer review?** For information about this choice, including consent withdrawal, please see our Privacy Policy .

Reviewer #1: **Yes: ** Fariba Zare

Reviewer #3: **Yes: ** LM Ho

- - - - -

---

## [Author Response · Author response to Decision Letter 2]

26 Jul 2024

RE manuscript PONE-D-23-23924R1

Preventing depression in high-income countries – a systematic review of studies evaluating change in social determinants

Dear editorial team,

Thank you for allowing is to resubmit our manuscript “Preventing depression in high-income countries – a systematic review of studies evaluating change in social determinants”.

As you are aware, this manuscript was recently rejected but is now being resubmitted after appeal. We would first like to summarise the facts. Initial submission was in July 2023 but took some time to be assigned to an editor. We received notification of the difficulties in assigning an editor on the 30th of October and then notification in November that an editor had been assigned. Reviewer comments were provided in February 2024, recommending revision of the manuscript, which we completed at the end of March. In June we received notification that the manuscript was again delayed due to changes in editors. Finally, on the 25th of June we received a rejection letter from the editor newly assigned to our paper.

The new editor names a number of reasons for rejection of our paper. However we would argue that some of the issues mentioned do not align for the overall aim and rationale for our review. Below we provide a response (in italics) to the comments made by the editor.

a) First, this manuscript only searches three journal databases, which may miss important information or references.

Three to four databases for systematic reviews are widely accepted in publications, also in other systematic reviews published in PLOS One. I can appreciate that 4-5 databases is more common in 2024 (also considering Google Scholar as a common database). If this was an issue it should have been flagged up in the first or second review of the article, when the authors could have chosen to search an additional database or take the review for consideration elsewhere. We discuss this in the paper and our strategy to overcome this shortcoming (page 83): “We conducted our search in three major databases (Medline, Embase and Psychinfo) so some relevant papers may have been missed. To compensate, we conducted a thorough search of the reference lists of included studies and published reviews”

b) The literature included by the author consists of quantitative and qualitative research. The author conducts the discussion using a qualitative method. Using qualitative research methods to perform a quantitative literature review will miss much information and is unscientific and unobjective.

Firstly, we only include quantitative research in our search, so this is a mistaken impression.

Secondly, our use of a narrative review is justified by the broad range of topics studied. Our aim was to provide a broad overview of the topic, including many determinants. A meta-analysis was not possible given the design of our review.

Narrative descriptions of evidence is not unscientific or unobjective, and we also rated the quality of the evidence included in the review, which is systematic and more objective.

c) The author excludes patients with diabetes and heart disease. What about other illnesses such as cancer, mental disorders, etc.? What is the basis for the exclusion?

As explained on page 8 of the manuscript, we excluded studies in specific clinical populations or all patient groups, of which persons with diabetes, heart disease are an example. Our aim is to give an overview of strategies that could be employed to prevent depression in the general population. We justify our exclusion of patient groups as the social determinants underlying depression risks in patients differ from these in de general population.

d) The authors exclude low-income and middle-income countries. Are these countries not providing social support? For example, China. When these countries are excluded, the conclusions of this manuscript will be biased.

We aimed to develop an overview of potential policies and intervention that act on social determinants of mental health relevant for high-income country contexts, and we acknowledge that we may have missed specific mechanisms at play in low and middle-income countries (page 83). However, the focus on high-income countries was based on the expectation that differences in context would have an influence on the relationship between social determinants and mental health. This does not imply that social determinants are not relevant for low and middle income countries, as the editor suggests. Understanding the relation between social determinants and mental health in those settings simply goes beyond the aim of this specific paper.

To further expand on this point in the manuscript we expanded somewhat on this point in the discussion, lines 874-880 where we added the text:

“Excluding low and middle-income countries in our search means that we might have missed capturing the mechanisms underlying the association between these determinants and depression. However, the influence of the social determinants can be very context-specific. For example, the level of economic development is likely to influence the impact of such interventions, especially progressive policies like parental leave, although there is likely to be variation in contextual factors between high-income countries as well”.

If this was an issue it should have been brought up in previous revisions of the systematic review, not already more than a year after in the review process of PLOS one.

e) In the abstract, the authors find evidence that strategies such as promoting paid employment and parental leave can reduce the risk of depression. What is the author's evidence? Is it through statistical analysis? I haven't seen any system evidence of this.

The evidence is in the included studies in our review showing this, but also here: How do income changes impact on mental health and wellbeing for working-age adults? A systematic review and meta-analysis (thelancet.com) and here: National or population level interventions addressing the social determinants of mental health – an umbrella review | BMC Public Health | Full Text (biomedcentral.com). This is not an opinion of the authors, but of the evidence synthesized in this review. The aim of a narrative systematic review is not do a statistical analysis. Our analysis is based on a rigorous evaluation of the evidence, which have been judged as ‘sound’ by the reviewers. In addition, it is not clear to us what the editor means with ‘any system evidence’?

f) Finally, using qualitative methods for the literature review of quantitative studies was permissible in the past when tools were not available. However, many software tools are now available to assist researchers in conducting more objective and scientific analyses. In particular, this manuscript includes many references, most of which are quantitative. Using traditional manual methods to organize and summarise would have missed much important information and would not have been objective.

We are of course aware of the availability of software tools to quantitatively summarize the results of multiple studies in a systematic review. However, the aim of our paper was NOT to quantitatively assess the strength of the association. Instead, the rationale of our study was to assess evidence for a causal relationship between social determinants and depression. As explained in line 81 and further: However, in existing studies it is often not clear whether a change in exposure to social determinants will also lead to a lower risk of depression. Establishing causality is fraught with difficulty and the increased application of counterfactual methodologies to this field has helped to clarify causality.

For this reason, the quantitative assessment as suggested by the editor goes beyond the aim of our paper.

g) In addition, the literature search and the inclusion criteria make this manuscript subject to serious bias and will miss important information or literature. It is recommended that the authors expand the literature search and adopt a quantitative method. Authors may include too wide a range of variables for quantitative methods can not be used. Authors are advised to focus on the research topic and expand the search of journal databases.

As explained in our previous reactions, this would entail a completely new approach, with a different research question.

We would like to point out that the views of the two reviewers that evaluated our (revised) manuscript were satisfied with the adjustments made. On initial evaluation of our manuscript one of the reviewers comments “The manuscript undertakes systematic review on a very pertinent topic highlighting whether changes in social determinants can prevent depression and its symptoms in high income countries”. Critical comments were made by both reviewers which we had responded to. Both reviewers answered gave a positive response to the question “is the manuscript technically sound, and do the data support the conclusions?”

We look forward to further consideration of this work.

With kind regards,

Mary Nicolaou, on behalf of the co-authors

---

## [Editor Report · Decision Letter 2]

8 Apr 2025

Preventing depression in high-income countries – a systematic review of studies evaluating change in social determinants

PONE-D-23-23924R2

Dear Dr. Nicolaou,

We’re pleased to inform you that your manuscript has been judged scientifically suitable for publication and will be formally accepted for publication once it meets all outstanding technical requirements.

I was informed of  the situation surrounding your submission. I am pleased with the editorial team's judicious decision to grant and accept your appeal.  Upon my close reading, I find this manuscript to be of great value. As you pointed out, there is not much review out there that focuses specifically on changes in exposure to social determinants. I acknowledge the authors' extraordinary efforts to provide future researchers with critical pointers to exploit pseudo experimental settings due to policy changes so comprehensively. The delay is indeed regrettable, but I would like to offer a strong endorsement  that this accomplished literature review is very much still relevant to ongoing research as of 2025.

With respect,

Katsuya oi, PhD

Academic Editor

PLOS ONE
---

## [Editor Report · Acceptance letter]

PONE-D-23-23924R2

PLOS ONE

Dear Dr. Nicolaou,

I'm pleased to inform you that your manuscript has been deemed suitable for publication in PLOS ONE. Congratulations! Your manuscript is now being handed over to our production team.

Kind regards,

on behalf of

Dr. katsuya oi

Academic Editor

PLOS ONE